# Learning Convolutional Representations via Generalized Stein's Method: A Training-Free Approach

## Abstract

Convolutional Neural Networks (CNNs) have revolutionized computer vision, with the convolution operation serving as a cornerstone that enables the extraction of abstract features and the discovery of hidden structures in image data. However, CNNs training typically relies on gradient descent, which can be computationally expensive and unstable, particularly in high-dimensional, small-sample settings such as medical imaging analysis. This paper presents an efficient statistical approach to learn convolutional representations without training a CNN. We reformulate CNNs into a general index model with matrix-valued inputs, interpreting convolution filters as index vectors while absorbing subsequent layers into the link function. Through a generalized version of the first-order Stein's formula, we develop a novel singular value decomposition (SVD) based approach to estimate the convolution filters directly. Theoretical analysis suggests that our estimation achieves an optimal convergence rate, comparable to that of generalized linear models where the link function is known. Extensive simulations and medical imaging experiments demonstrate the effectiveness of our approach, providing a viable pathway for representation learning.

## 1 Introduction

In modern machine learning, representation learning has emerged as a fundamental paradigm that focuses on automatically discovering meaningful and compact features from raw data. With the popularity of deep learning, representation learning has made significant advances across diverse fields, including computer vision (Sharif Razavian et al., 2014; Dosovitskiy et al., 2020; Liu et al., 2024), natural language processing (Vaswani, 2017; Singh et al., 2022; Patil et al., 2023) and more. Owing to the sophisticated architectures of these models, deep learning-based representations are capable of capturing latent and complex patterns in the data, enabling improved performance on various downstream tasks such as classification, prediction, etc.

As one of the most influential frameworks in image analysis, Convolutional Neural Networks (CNNs) exemplify modern representation learning. Their advantage lies in the nonlinearity and convolutional representations, which preserve and exploit rich spatial structure of images—an aspect that classical dimension-reduction methods like principal component analysis (PCA) or linear discriminant analysis (LDA) struggle to capture. These learned representations make CNNs powerful not only for prediction but also as feature extractors, with convolutional layers offering features for subsequent tasks. Nevertheless, the convolution filters as the key unknown parameters of CNNs, are typically learned with gradient descent algorithms, which are computationally expensive and face stability issues, especially in the high-dimensional but small-sample settings like medical imaging analysis. This motivates the exploration of alternative representation learning approaches that retain structural information while enjoying greater efficiency.

Serving as a bridge between dimension reduction and nonlinearity, the index model is a classical statistical framework that generalizes linear regression to a nonlinear format. Specifically, a multi-index model (MIM) usually takes the form of $\mathbb{E}(y) = f(\boldsymbol{x}^\top \boldsymbol{b}_1, \ldots, \boldsymbol{x}^\top \boldsymbol{b}_R)$, where $\boldsymbol{b}_1, \ldots, \boldsymbol{b}_R$ are the target index vectors, and $f$ is an unknown multi-variate nonlinear function, referred to as the link function. When $R = 1$, the MIM reduces to the single-index model (SIM), which leads to a single

index $\boldsymbol{x}^\top \boldsymbol{b}$ and a univariate link function $f$. Index models not only offer considerable adaptability by incorporating the nonlinear link function, but also avoid the curse of dimensionality through a low-dimensional representation of the input.

Motivated by feature extraction capabilities of CNNs and the objective for greater efficiency and stability, this paper presents a novel approach for learning convolutional representations in medical imaging data. We begin with formulating CNNs into a general index model with matrix-valued inputs, where convolution filters are treated as index vectors while the subsequent pooling or fully-connected layers are absorbed into the unknown link function. Leveraging a generalized first-order Stein's formula, we propose to estimate convolutional filters through singular value decomposition (SVD), which is thus efficient and training-free in the sense that it does not require training a CNN. Theoretical analysis suggests that our estimation achieves an optimal convergence rate, matching that of generalized linear models where the link function is known. We conduct extensive simulation studies and real data analyses and the results show that our approach even outperforms popular deep learning algorithms. To summarize, the key contributions of this work are:

- Formulate CNNs into a general index model with matrix-valued inputs, and propose a novel SVD-based convolutional representation learning approach using Stein's formula.

- Establish theoretical guarantees showing that our estimation achieves an optimal convergence rate, comparable to that of generalized linear models with a known link function.

- Comprehensive simulations and real data experiments demonstrate the effectiveness of our convolutional representation learning approach, improving both performance and computational efficiency over popular deep learning algorithms.

**Related works.** Given that the link function $f$ is unknown, directly estimating the index vectors through least squares is generally impractical. For the simplest case where $\boldsymbol{x}$ is a standard Gaussian random vector, Stein's formula (Stein, 1981) indicates that the index vector $\boldsymbol{b}$ is proportional to $\mathbb{E}(\boldsymbol{x} \cdot y)$, regardless of the link function $f$. A recent line of research, mostly centered on single index models, suggests using the moment-based method for index vector estimation when the distribution of $\boldsymbol{x}$ is known. Based on Stein's method and its generalized version, various extensions have been considered in the literature. For instance, Plan and Vershynin (2016); Plan et al. (2017) considered SIM with sparse index vectors. Yang et al. (2017a); Goldstein et al. (2018) extended the Gaussian input assumption to heavy-tailed or non-Gaussian scenarios. Fan et al. (2023) further studied implicit regularization in single index models. Compared to SIM, research on multi-index models is much more limited. Initial groundbreaking works by Ker-Chau Li, including regression under link violation (Li and Duan, 1989), sliced inverse regression (Li, 1991), and principal Hessian directions (Li, 1992), paved the way for MIM. More recently, Yang et al. (2017b) proposed learning non-Gaussian MIM through a second-order Stein's Method. Beyond MIM, the Stein's Method has also been applied to other models, such as varying coefficient index model (Na et al., 2019).

In the machine learning community, index models have also been considered as the target to evaluate the abilities of neural networks in learning low-dimensional representations. For example, Mousavi-Hosseini et al. (2022) studied the problem of training a two-layer neural network via stochastic gradient descent when the target follows a multi-index model. Additionally, Bietti et al. (2022) considered a class of shallow neural networks and investigated its ability in learning single-index models via gradient flow.

**Notations.** We use bold uppercase letters $\boldsymbol{A}$, $\boldsymbol{B}$ to denote matrices, bold lowercase letters $\boldsymbol{a}$, $\boldsymbol{b}$ to denote vectors. We let $\mathrm{vec}(\cdot)$ be the vectorization operator and $\mathrm{vec}_{(\cdot)}^{-1}(\cdot)$ be its inverse with the subscripts indicating the matrix size. For example, $\mathrm{vec}_{(p,q)}^{-1}(\cdot)$ stands for transforming a vector of dimension $pq$ to a matrix of dimension $p \times q$. For a vector $\mathbf{v}$, $\|\mathbf{v}\|_q = (\sum_j |v_j|^q)^{1/q}$ is the $\ell_q$ norm. For a matrix $\boldsymbol{A}$, $\|\boldsymbol{A}\|_F = \sqrt{\sum_{i,j} \boldsymbol{A}_{i,j}^2}$ is the Frobenius norm, $\lambda_{min}(\boldsymbol{A})$ is the minimum eigenvalue of matrix $\boldsymbol{A}$. Moreover, we let $\langle \cdot, \cdot \rangle$ to denote the inner product. In addition, given two sequences $\{x_n\}$ and $\{y_n\}$, we denote $x_n = \mathcal{O}(y_n)$ if $|x_n| \leqslant C_1 |y_n|$ for some absolute constant $C_1$ and denote $x_n = \Omega(y_n)$ if $|x_n| \geqslant C_2 |y_n|$ for some absolute constant $C_2$.

## 2 CNN, Index Model, and their connections

### 2.1 CNN and Index Model

Let $\boldsymbol{X}^{\mathrm{ori}} \in \mathbb{R}^{P_1 \times P_2}$ denote the original matrix-valued input, such as raw image data. In a typical convolutional neural network, the raw image usually first goes through a convolutional layer, and then proceeds to pooling and fully connected layers. We shall focus on the first convolutional layer and denote $\boldsymbol{B}_1, \ldots, \boldsymbol{B}_R \in \mathbb{R}^{d_1 \times d_2}$ as $R$ convolution filters in the first layer of a CNN. Then, the output of a convolutional neural network can be written as the following form

$$G(\boldsymbol{X}^{\mathrm{ori}}) = g(\sigma(\boldsymbol{X}^{\mathrm{ori}} \star \boldsymbol{B}_1), \ldots, \sigma(\boldsymbol{X}^{\mathrm{ori}} \star \boldsymbol{B}_R)). \tag{1}$$

Here $\star$ is a convolution operator to be discussed in detail later, $\sigma(\cdot)$ is a component-wise nonlinear activation function, such as ReLU (Rectified Linear Unit) or Sigmoid function. Furthermore, the multivariate function $g$ incorporates the remaining structure of CNN, including pooling layers, fully-connected layers, and other components in a CNN. If we further combine $g$ with the activation function $\sigma$, the CNN output (1) can be written as

$$G(\boldsymbol{X}^{\mathrm{ori}}) = f(\boldsymbol{X}^{\mathrm{ori}} \star \boldsymbol{B}_1, \ldots, \boldsymbol{X}^{\mathrm{ori}} \star \boldsymbol{B}_R). \tag{2}$$

We now turn our attention to the convolution operator $\star$. We shall first consider a non-overlapping convolution, further extensions to its overlapping version will be discussed later. Given any matrix $\boldsymbol{M} \in \mathbb{R}^{P_1 \times P_2}$ and a filter $\boldsymbol{B} \in \mathbb{R}^{d_1 \times d_2}$, where $d_1, d_2$ are factors of $P_1, P_2$ respectively, the non-overlapping convolution is defined as

$$\boldsymbol{M} \star \boldsymbol{B} \in \mathbb{R}^{p_1 \times p_2}, \quad (\boldsymbol{M} \star \boldsymbol{B})_{j,k} = \left\langle \boldsymbol{M}_{j,k}^{d_1, d_2}, \boldsymbol{B} \right\rangle. \tag{3}$$

Here, $(p_1, p_2) = (P_1/d_1, P_2/d_2)$ and $\boldsymbol{M}_{j,k}^{d_1, d_2}$ represents the $(j, k)$-th block of size $d_1 \times d_2$ in $\boldsymbol{M}$ for $1 \leqslant j \leqslant p_1$ and $1 \leqslant k \leqslant p_2$. With deep learning language, the stride size is set equal to the filter size under non-overlapping scenario. The convolution operation aims to extract feature information, thereby enhancing prediction in the subsequent layers. While through the non-overlapping design, dimension reduction can be achieved. With non-overlapping convolution, the function $f$ in (2) is a multivariate function with the input of dimension $p_1 \times p_2 \times R$. Notably, the non-overlapping convolution, being the simplest case, has been extensively studied in the literature, for example, Brutzkus and Globerson (2017); Du et al. (2018); Cao and Gu (2019); Feng and Yang (2024).

Now we consider a matrix reshaping operator that allows us to further connect the CNN and index models. Given any matrix $\boldsymbol{M} \in \mathbb{R}^{P_1 \times P_2}$, the operator $\mathcal{R}_{(d_1, d_2)} : \mathbb{R}^{P_1 \times P_2} \to \mathbb{R}^{(p_1 p_2) \times (d_1 d_2)}$ is defined as a mapping from $\boldsymbol{M}$ to

$$\mathcal{R}_{(d_1, d_2)}(\boldsymbol{M}) = \left[ \mathrm{vec}\left( \boldsymbol{M}_{1,1}^{d_1, d_2} \right), \ldots, \mathrm{vec}\left( \boldsymbol{M}_{p_1, p_2}^{d_1, d_2} \right) \right]^{\top}. \tag{4}$$

A key property holds for $\boldsymbol{X}^{\mathrm{ori}} \in \mathbb{R}^{P_1 \times P_2}$ and $\boldsymbol{B} \in \mathbb{R}^{d_1 \times d_2}$ that

$$\mathrm{vec}(\boldsymbol{X}^{\mathrm{ori}} \star \boldsymbol{B}) = \mathcal{R}_{(d_1, d_2)}(\boldsymbol{X}^{\mathrm{ori}})\mathrm{vec}(\boldsymbol{B}). \tag{5}$$

Let $\boldsymbol{X} = \mathcal{R}(\boldsymbol{X}^{\mathrm{ori}}) \in \mathbb{R}^{(p_1 p_2) \times (d_1 d_2)}$ be the reshaped input and $\boldsymbol{b}_r = \mathrm{vec}(\boldsymbol{B}_r) \in \mathbb{R}^{(d_1 d_2)}$ for $r = 1, \ldots, R$. Then we can rewrite (2) as

$$G(\boldsymbol{X}^{\mathrm{ori}}) = f(\boldsymbol{X}\boldsymbol{b}_1, \ldots, \boldsymbol{X}\boldsymbol{b}_R). \tag{6}$$

With slight abuse of notation, we also use the same function notation $f$ in (6). However, we shall note that the $f$ in (6) is not identical to that in (2) as the inputs were reshaped. If we model certain outcome $Y$ through above formulation, i.e.,

$$\mathbb{E}(Y) = f(\underbrace{\boldsymbol{X}\boldsymbol{b}_1, \ldots, \boldsymbol{X}\boldsymbol{b}_R}_{(p_1 p_2) \times R}). \tag{7}$$

The model (7) clearly bears a resemblance to an MIM, although in fact they are different. In an MIM, the input is typically a vector, which results in each index being a scalar. In (7), however, $\boldsymbol{X}$ is a matrix of size $(p_1 p_2) \times (d_1 d_2)$, which makes each index $\boldsymbol{X}\boldsymbol{b}_r$ a vector with a dimension of $p_1 p_2$. As a result, the function in (7) is a much more general function with an input dimension of $p_1 p_2 \times R$.

Furthermore, let $p = p_1 p_2$ be the total number of non-overlapping blocks and $d = d_1 d_2$ be the number of components in each block. Moreover, let $\boldsymbol{\Theta} = (\boldsymbol{b}_1, \ldots, \boldsymbol{b}_R) \in \mathbb{R}^{d \times R}$ be the collection of convolution filters. In the next section, we introduce the approach to estimate $\boldsymbol{\Theta}$.

## 2.2 CNN FILTER ESTIMATION VIA STEIN'S METHOD AND SVD

Let $\boldsymbol{X}_i = \mathcal{R}(\boldsymbol{X}_i^{\mathrm{ori}}) \in \mathbb{R}^{p \times d}$ and $Y_i \in \mathbb{R}$ for $i \in [n]$ be $n$ i.i.d. observations that follow model (7), i.e., $\mathbb{E}(Y_i) = f(\boldsymbol{X}_i \boldsymbol{\Theta})$. In this section, we introduce a novel approach to estimate $\boldsymbol{\Theta} = (\boldsymbol{b}_1, \ldots, \boldsymbol{b}_R) \in \mathbb{R}^{d \times R}$ without knowing $f(\cdot)$. Similar to index models, $\boldsymbol{\Theta}$ is not identifiable when $f$ is unknown. Therefore, our goal is to learn the $R$ column space in $\boldsymbol{\Theta}$ based on a generalized version of Stein's identity. To get started, we shall first define the score function associated with the input $\boldsymbol{X}$.

**Definition 2.1.** *Let* $\boldsymbol{X} \in \mathbb{R}^{p \times d}$ *be a random matrix with density* $P : \mathbb{R}^{p \times d} \to \mathbb{R}$. *The score function* $S(\boldsymbol{X}) : \mathbb{R}^{p \times d} \to \mathbb{R}^{p \times d}$ *associated with* $\boldsymbol{X}$ *is defined as*

$$S(\boldsymbol{X}) := -\nabla_{\boldsymbol{X}}[\log P(\boldsymbol{X})] = -\nabla_{\boldsymbol{X}} P(\boldsymbol{X}) / P(\boldsymbol{X}).$$

*For any* $\{j, k\} \in [p] \times [d]$, $S_{jk}(\boldsymbol{X})$ *is defined as the* $(j, k)$-*th element of* $S(\boldsymbol{X})$.

Then we have the following generalized version of first-order Stein's identity.

**Lemma 2.2.** *Let* $f : \mathbb{R}^{(p_1 p_2) \times R} \mapsto \mathbb{R}$ *be the link function. Let* $\boldsymbol{Z} = \boldsymbol{X} \boldsymbol{\Theta}$. *Suppose that the expectations* $\mathbb{E}[Y S(\boldsymbol{X})]$ *and* $\mathbb{E}[\nabla_{\boldsymbol{Z}} f(\boldsymbol{X} \boldsymbol{\Theta})]$ *both exist and are well-defined. Then we have*

$$\mathbb{E}[Y S(\boldsymbol{X})] = \mathbb{E}[\nabla_{\boldsymbol{Z}} f(\boldsymbol{X} \boldsymbol{\Theta})] \cdot \boldsymbol{\Theta}^{\top}. \tag{8}$$

Lemma 2.2 serves as the basis for our estimation of $\boldsymbol{\Theta}$. By Lemma 2.2, regardless of the specific form of link function $f$, the column space of $\boldsymbol{\Theta}$ can be determined by the top $R$ right-singular vectors of $\mathbb{E}[Y S(\boldsymbol{X})]$. With the sample version $(1/n) \sum_{i=1}^{n} Y_i S(\boldsymbol{X}_i)$ being a natural estimate for $\mathbb{E}[Y S(\boldsymbol{X})]$, we propose the following SVD based estimation for $\boldsymbol{\Theta}$,

$$\widehat{\boldsymbol{\Theta}} = \mathrm{SVD}_{v,R}\left(\frac{1}{n} \sum_{i=1}^{n} Y_i S(\boldsymbol{X}_i)\right), \tag{9}$$

where $\mathrm{SVD}_{v,R}(\boldsymbol{M})$ refers to the top-R right-singular vectors of a matrix $\boldsymbol{M}$.

We shall emphasize that our estimator does not rely on the link function $f$. This implies that a direct estimation of the convolution filter can be accomplished regardless of the complexity of the pooling or fully-connected layer in a CNN. This estimation is general in the sense that it works for an arbitrary distribution of input $\boldsymbol{X}$, provided that the score $S(\boldsymbol{X})$ is known. Furthermore, if the density of $\boldsymbol{X}$ is unknown, a plug-in estimation for $S(\boldsymbol{X})$ can be considered for any parametric form of density. Additional details regarding unknown densities will be discussed in Section 4.

# 3 THEORETICAL GUARANTEES

## 3.1 THEORETICAL GUARANTEES FOR SINGLE FILTER ESTIMATION

We first provide theoretical guarantees for estimating $\boldsymbol{\Theta}$ with a single convolution filter. With $R = 1$, let $\boldsymbol{b} = \mathrm{vec}(\boldsymbol{B}) = \boldsymbol{\Theta} \in \mathbb{R}^d$ be the target filter. Denote the estimated filter as $\widehat{\boldsymbol{b}} = \widehat{\boldsymbol{\Theta}}$. In other words,

$$\widehat{\boldsymbol{b}} = \mathrm{SVD}_{v,1}\left(\frac{1}{n} \sum_{i=1}^{n} Y_i S(\boldsymbol{X}_i)\right).$$

Throughout this section, we assume that $\|\boldsymbol{b}\|_2 = 1$ to avoid identifiability issues. We aim to prove that $\|\widehat{\boldsymbol{b}} - \boldsymbol{b}\|_2$ is rate-optimal up to a logarithmic factor under rather general assumptions. To get started, we first present the necessary conditions.

**Assumption 3.1.** *Suppose that* $\boldsymbol{X}_i \in \mathbb{R}^{p \times d}$, $i \in [n]$ *are i.i.d. observations.*

(a) *For any* $i \in [n]$, $j \in [p]$ *and* $k \in [d]$, *assume that the score* $S_{jk}(\boldsymbol{X}_i)$ *is a sub-Gaussian random variable, i.e.* $(\mathbb{E}|S_{jk}(\boldsymbol{X}_i)|^q)^{\frac{1}{q}} \leqslant L\sqrt{k}$ *for* $q \geqslant 1$, *where* $L$ *is a positive constant.*

(b) *Assume that there exists a constant* $T > 0$ *such that* $\mathbb{E}[f(\boldsymbol{X} \boldsymbol{\Theta})^{2q}]^{\frac{1}{2q}} \leqslant T$ *for all* $q \in \mathbb{N}_+$.

(c) *Let* $\boldsymbol{Z} = \boldsymbol{X} \boldsymbol{b}$. *For any* $j \in [p]$, *assume that* $\mathbb{E}[\nabla_{Z_j} f(\boldsymbol{Z})] = \Omega(1)$.

Assumption 3.1(a) requires that each entry of the score $S(\boldsymbol{X})$ is sub-Gaussian random variable. This assumption is rather mild and can be satisfied by a wide range of random distributions. Furthermore, we note that our analyses can be extended to general scores without the sub-Gaussian assumption on the score. We defer to Section B.1 for the detailed analysis.

Assumption 3.1(b) requires that the expectation of $\left[f(\boldsymbol{X\Theta})^{2q}\right]^{\frac{1}{2q}}$ is upper bounded. It automatically holds when $f \in \mathbb{R}^{(p_1 p_2) \times R} \mapsto \mathbb{R}$ is a bounded function. Moreover, when $f$ is Lipschitz continuous and $\boldsymbol{X}$ is sub-Gaussian, it is easy to show that Assumption 3.1(b) could also be satisfied.

In Assumption 3.1(c), we assume that the derivative of the non-linear function $f$ at $\boldsymbol{Z} = \boldsymbol{X b}$ maintains a constant level. This assumption ensures identifiability by requiring that the function value $f(\boldsymbol{X b})$ is adequately sensitive to perturbations around the true value $\boldsymbol{b}$. This assumption is mild, as it merely assumes a constant scale of the derivatives in expectation.

**Theorem 3.2.** *Suppose the model $\mathbb{E}(Y_i) = f(\boldsymbol{X}_i \boldsymbol{b})$ for $i \in [n]$. Under Assumption 3.1, we have with probability at least $1 - \delta$ that*

$$\|\widehat{\boldsymbol{b}} - \boldsymbol{b}\|_2 = \mathcal{O}\left(\sqrt{\frac{d}{n} \cdot \log \frac{pd}{\delta}}\right). \tag{10}$$

Theorem 3.2 suggests that the $\ell_2$-convergence rate of $\widehat{\boldsymbol{b}}$ is $\mathcal{O}\left(\sqrt{d \log(pd)/n}\right)$ under the sub-Gaussian setting. This rate is optimal up to a logarithmic factor. In fact, consider the simplest scenario where $f$ is linear, meaning that $\mathbb{E}(Y_i) = \boldsymbol{c}^\top \boldsymbol{X}_i \boldsymbol{b}$, with $\boldsymbol{c} \in \mathbb{R}^p$ being a known vector. In this case, the $\ell_2$-convergence rate of an OLS estimate of $\boldsymbol{b}$, denoted as $\widehat{\boldsymbol{b}}^{ols}$, is $\mathcal{O}\left(\sqrt{d/n}\right)$. Our rate in Theorem 3.2 matches with this OLS rate apart from a logarithmic factor, which confirms the optimality of (10).

## 3.2 Generalization to multiple filters estimation

In this section, we consider general cases with $R \geqslant 2$ filters in CNNs. However, as discussed, when $R \geqslant 2$, $\boldsymbol{\Theta}$ is not identifiable with an unknown link function $f$. Thus, our goal is to learn the column space of $\boldsymbol{\Theta}$ and prove the convergence of the estimators using the following distance measure

$$\text{dist}\left(\boldsymbol{\Theta}, \widehat{\boldsymbol{\Theta}}\right) = \inf_{\boldsymbol{H} \in \mathbb{H}_R} \|\boldsymbol{\Theta} - \widehat{\boldsymbol{\Theta}} \boldsymbol{H}\|_F, \tag{11}$$

where $\mathbb{H}_R$ is the set of $R \times R$ orthogonal matrices. It is worth noting that when $R = 1$, the column space distance $\text{dist}\left(\boldsymbol{b}, \widehat{\boldsymbol{b}}\right)$ reduces to the vector distance $\|\widehat{\boldsymbol{b}} - \boldsymbol{b}\|_2$ when $\|\boldsymbol{b}\|_2 = \|\widehat{\boldsymbol{b}}\|_2$.

In the context of a multi-filter situation, we require a similar set of assumptions as discussed in Section 3.1. The only deviation is that the bounded derivatives Assumption 3.1(c) should be replaced by Assumption 3.3 presented below.

**Assumption 3.3.** *Let $\sigma_R$ be the $R$-th largest singular value of $\mathbb{E}[Y_i S(\boldsymbol{X}_i)]$. Assume that there exists an absolute constant $C$ such that $\sigma_R \geqslant C\sqrt{p/R}$.*

Assumption 3.3 imposes a lower bound of the non-zero singular values of $\mathbb{E}[Y_i S(\boldsymbol{X}_i)]$. This assumption can be viewed as an extension of Assumption 3.1(c) in a multiple filters scenario. For example, when each entry of $\mathbb{E}[\nabla_{\boldsymbol{Z}} f(\boldsymbol{X\Theta})]$ is $\Omega(1/\sqrt{R})$ and the columns of $\mathbb{E}[\nabla_{\boldsymbol{Z}} f(\boldsymbol{X\Theta})]$ are nearly orthogonal, it is easy to show that all the $R$ non-zero singular values of $\mathbb{E}[Y S(\boldsymbol{X})]$ have a lower bound in scale of $\Omega(\sqrt{p/R})$.

**Theorem 3.4.** *Suppose the model $\mathbb{E}(Y_i) = f(\boldsymbol{X}_i \boldsymbol{\Theta})$ for $i \in [n]$. Under Assumption 3.1(a), (b) and Assumption 3.3, with probability at least $1 - \delta$, we have*

$$\inf_{\boldsymbol{H} \in \mathbb{H}_R} \|\widehat{\boldsymbol{\Theta}} \boldsymbol{H} - \boldsymbol{\Theta}\|_F = \mathcal{O}\left(\sqrt{\frac{Rd}{n} \cdot \log \frac{pd}{\delta}}\right).$$

In this multiple filters case, we achieve a convergence rate of $\mathcal{O}\left(\sqrt{Rd \log(pd)/n}\right)$ under the sub-Gaussian assumption. By replacing Assumption 3.1(c) with an SVD-type Assumption 3.3, Theorem 3.4 extends the near-optimal convergence rate stated in Theorem 3.2 to a multiple filter scenario.

## 4 GAUSSIAN DISTRIBUTION WITH UNKNOWN COVARIANCES

To estimate convolution filters using our approach, a crucial step involves computing the score function, which requires knowledge of the input distribution. Nevertheless, the input distributions are usually unknown in practice. When the input distribution has a specific parametric form, a natural solution is to use the plug-in estimator. This involves estimating the unknown parameters from the samples and then plugging them into the scores. In this section, we take the Gaussian input as an example and demonstrate the convergence of the plug-in estimators.

Let $\boldsymbol{x}_i = \text{vec}(\boldsymbol{X}_i) = \text{vec}\left(\mathcal{R}\left(\boldsymbol{X}_i^{\text{ori}}\right)\right)$ be the vectorized image. Assume that $\boldsymbol{x}_i \sim \mathcal{N}(\boldsymbol{\mu}, \boldsymbol{\Sigma})$ follows the Gaussian distribution with unknown mean $\boldsymbol{\mu}$ and covariance matrix $\boldsymbol{\Sigma}$. In such a Gaussian case, it is easy to show that the score function reduces to $S(\boldsymbol{X}_i) = \text{vec}_{(p,d)}^{-1}(\boldsymbol{\Sigma}^{-1}\boldsymbol{x}_i)$. Denote the sample mean and sample covariance matrix as $\hat{\boldsymbol{\mu}} = (1/n)\sum_{i=1}^{n}\boldsymbol{x}_i$ and $\widehat{\boldsymbol{\Sigma}} = (1/n)\sum_{i=1}^{n}\{\boldsymbol{x}_i - \hat{\boldsymbol{\mu}}\}\{\boldsymbol{x}_i - \hat{\boldsymbol{\mu}}\}^{\top}$, respectively. By plugging them into the score, we obtain the following estimator

$$\breve{\boldsymbol{\Theta}} = \text{SVD}_{v,R}\left(\breve{\boldsymbol{A}}\right), \quad \breve{\boldsymbol{A}} = \text{vec}_{(p,d)}^{-1}\left(\frac{1}{n}\sum_{i=1}^{n} Y_i \widehat{\boldsymbol{\Sigma}}^{-1}\left(\boldsymbol{x}_i - \hat{\boldsymbol{\mu}}\right)\right). \tag{12}$$

It can be proved that column space of the plug-in estimator converges to that of the original estimator with the rate $\mathcal{O}\left(\sqrt{Rd/n}\right)$, as stated in Theorem D.2 in the appendix. In fact, under certain cases, the plug-in estimator could even be advantageous over the original version. For example, consider a straightforward case where $\boldsymbol{X}_i$ is Gaussian with zero mean and $f(\cdot)$ is linear: $Y_i = \langle \boldsymbol{C}, \boldsymbol{X}_i\boldsymbol{\Theta}\rangle + \epsilon_i$. Here, $\boldsymbol{C} \in \mathbb{R}^{p \times R}$ represents a certain unknown matrix. Clearly, such a model could be viewed as a low-rank trace regression $\mathbb{E}(Y_i) = \text{tr}(\boldsymbol{A}\boldsymbol{X}_i^{\top})$, where $\boldsymbol{A} = \boldsymbol{C}\boldsymbol{\Theta}^{\top} \in \mathbb{R}^{p \times d}$ is a low-rank matrix. Under such a setting, $\breve{\boldsymbol{A}}$ in (12) is essentially the OLS estimate of $\boldsymbol{A}$

$$\breve{\boldsymbol{A}} = \text{vec}_{(p,d)}^{-1}\left(\frac{1}{n}\sum_{i=1}^{n}\widehat{\boldsymbol{\Sigma}}^{-1}\boldsymbol{x}_i Y_i\right) = \boldsymbol{A} + \text{vec}_{(p,d)}^{-1}\left(\frac{1}{n}\sum_{i=1}^{n}\widehat{\boldsymbol{\Sigma}}^{-1}\boldsymbol{x}_i \epsilon_i\right).$$

As a comparison, the original estimator $\widehat{\boldsymbol{A}}$ with true mean and covariance becomes

$$\widehat{\boldsymbol{A}} = \text{vec}_{(p,d)}^{-1}\left(\frac{1}{n}\sum_{i=1}^{n}\boldsymbol{\Sigma}^{-1}\boldsymbol{x}_i Y_i\right) = \text{vec}_{(p,d)}^{-1}\left(\boldsymbol{\Sigma}^{-1}\widehat{\boldsymbol{\Sigma}}\text{vec}\left(\boldsymbol{A}\right)\right) + \text{vec}_{(p,d)}^{-1}\left(\frac{1}{n}\sum_{i=1}^{n}\boldsymbol{\Sigma}^{-1}\boldsymbol{x}_i \epsilon_i\right).$$

By the above derivation, $\breve{\boldsymbol{A}}$ could potentially provide a better estimation of $\boldsymbol{A}$ compared to $\widehat{\boldsymbol{A}}$. Particularly, when sample size $n$ is small, $\boldsymbol{\Sigma}^{-1}\widehat{\boldsymbol{\Sigma}}$ might deviate from $\boldsymbol{I}$ and then the estimation performance of $\widehat{\boldsymbol{A}}$ could be impacted. Consequently, the plug-in estimator $\breve{\boldsymbol{\Theta}}$, which consists of the singular vectors of $\breve{\boldsymbol{A}}$, might yield a smaller estimation error. This finding is further validated by an extensive simulation study presented in Section 5.2.

## 5 SIMULATION STUDIES

In this section, we conduct a comprehensive simulation study to demonstrate the performance of the proposed estimator. We consider the following model with $R = 3$ filters

$$y_i = f(\boldsymbol{X}_i^{\text{ori}} \star \boldsymbol{B}_1, \ \boldsymbol{X}_i^{\text{ori}} \star \boldsymbol{B}_2, \ \boldsymbol{X}_i^{\text{ori}} \star \boldsymbol{B}_3) + \varepsilon_i. \tag{13}$$

The input images $\boldsymbol{X}_i^{\text{ori}}$ are of size $28 \times 28$ and the filters $(\boldsymbol{B}_1, \boldsymbol{B}_2, \boldsymbol{B}_3)$ are of size $4 \times 4$. Upon reshaping the image, we obtain $\boldsymbol{X}_i = \mathcal{R}_{(4,4)}(\boldsymbol{X}^{\text{ori}}) \in \mathbb{R}^{49 \times 16}$. We consider different sample sizes with $n$ ranging from 500 to 5000. In addition, the noise term $\varepsilon_i$ is generated by Gaussian distribution $\mathcal{N}(0, \sigma^2)$ with $\sigma = 0.1$. We fix the values of the filters but vary the distributions of input $\boldsymbol{X}_i^{\text{ori}}$ and link function $f$. Specifically, we consider $\boldsymbol{X}^{\text{ori}}$ of the following distributions.

(1) $\boldsymbol{X}^{\text{ori}}$ with i.i.d. Gaussian entries, $\boldsymbol{X}_{jk}^{\text{ori}} \sim \mathcal{N}(\mu, \sigma^2)$, $\mu = 0$ and $\sigma = 1$.

(2) $\boldsymbol{X}^{\text{ori}}$ with correlated Gaussian entries. Specifically, we generate $\boldsymbol{X}^{\text{ori}}$ by $\text{vec}(\boldsymbol{X}_i^{\text{ori}}) \sim \mathcal{N}(\boldsymbol{\mu}, \boldsymbol{\Sigma})$. The mean vector $\boldsymbol{\mu}$ has uniformly random integer entries ranging from $[-5, 5]$, while the covariance matrix $\boldsymbol{\Sigma}$ has entries $\boldsymbol{\Sigma}_{j,k} = \rho^{|j-k|}$, with different values of $\rho$.

Furthermore, we consider the following types of link function $f$.

(I) Linear: $f(\boldsymbol{Z}) = \langle \boldsymbol{Z}, \boldsymbol{C} \rangle$, where the coefficient $\boldsymbol{C}$ has i.i.d. Gaussian entries $\boldsymbol{C}_{jk} \sim \mathcal{N}(0, 1)$.

(II) Nonlinear: $f(\boldsymbol{Z}) = \langle \boldsymbol{Z} + 3\sin(\boldsymbol{Z}), \boldsymbol{C} \rangle$, where $\boldsymbol{C}$ is defined as in (I). Note that here the function $\sin(\boldsymbol{Z})$ is applied element-wisely.

(III) Fully connected network (FCN): $f(\boldsymbol{Z})$ is a two-layer fully connected network.

(IV) Convolutional neural network: $f(\boldsymbol{Z})$ is a convolutional neural network.

We defer to Section C.1 for a detailed description of the structure of the neural networks and additional details on this simulation study.

## 5.1 COMPARISON TO GRADIENT DESCENT

In this subsection, we evaluate the convolution filter estimation performance of our approach in comparison to CNNs optimized using Adam, implemented in *PyTorch*. The evaluation is based on the column distance metric discussed in (11). Here we shall focus on the original version of the estimator with known input distributions. Further discussions regarding unknown density estimation and the truncated estimator will be addressed later.

We emphasize again that our approach does not require the knowledge of link functions to learn the convolution filters. However, to learn these filters using Adam, the neural network structure must be specified. We consider a two-layer fully connected network to approximate the link functions (I) to (IV). Additionally, for DNN-based link functions (III) and (IV), we also consider a neural network that matches the true structure. Such an "oracle" specification evidently provides an advantage for Adam in learning the convolutions. Figure 1 below illustrates the estimation performance of our method and Adam for multiple-filter case. These results are based on 100 independent repetitions.

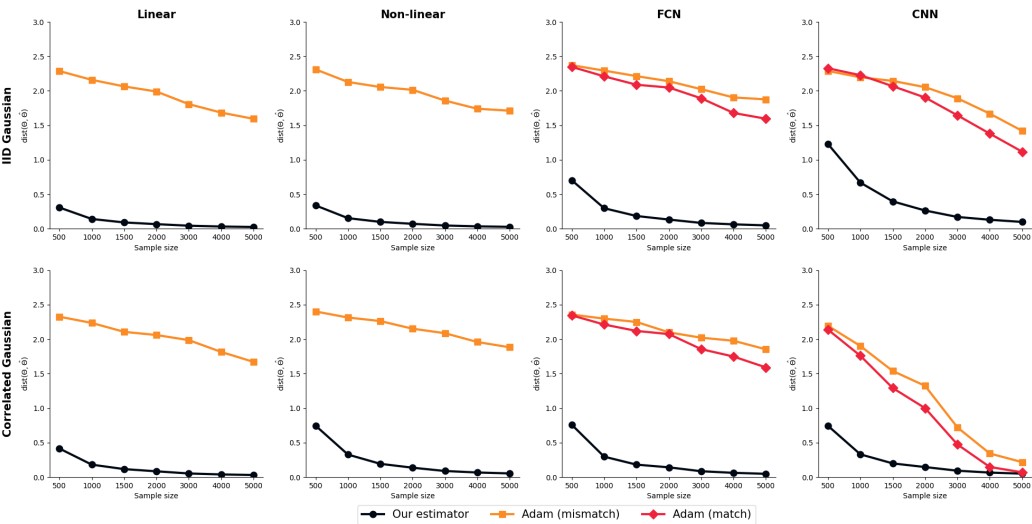

Figure 1: Convolution filter estimation performance of proposed approach and Adam under different link functions and input distributions.

By Figure 1, it is clear that our approach consistently performs well and dominates Adam across different link functions and input distributions in both single filter and multiple filters scenarios. In particular, even for the matched case where the neural network structures are correctly specified in cases (III) and (IV), our approach still exhibits significant advantages, particularly when the sample size $n$ is relatively small. As the sample size increases, the performance of the gradient-based method improves, leading to a decrease in filter estimation error. In fact, neural network training typically requires a large sample size to achieve the desired performance. However, in many instances, such as medical imaging analysis, the available sample sizes are often limited. In such situations, our approach, which has a much lower dependency on sample sizes, would be highly appreciated.

## 5.2 Unknown Gaussian distribution

In this subsection, we consider the scenario that the input distribution is unknown. We restrict our attention to Gaussian distributions with unknown means and covariance matrices to illustrate the performance of the plug-in estimator. In this experiment, we focus on the case (4), where the input $\boldsymbol{X}^{\mathrm{ori}}$ has correlated Gaussian entries: $\boldsymbol{X}^{\mathrm{ori}} \sim \mathcal{N}(\mu, \boldsymbol{\Sigma})$, with covariance matrices $\boldsymbol{\Sigma}_{i,j} = \rho^{|i-j|}$. We consider two different values of $\rho$, specifically $\rho = 0.5$ and $\rho = 0.8$, representing varying degrees of correlation. We evaluate the performance of the plug-in estimator, introduced in Section 4, by comparing it with the original estimator, where the mean and covariance matrix are correctly specified. Figure 2 below plots the estimation errors of two estimators under different $\rho$'s.

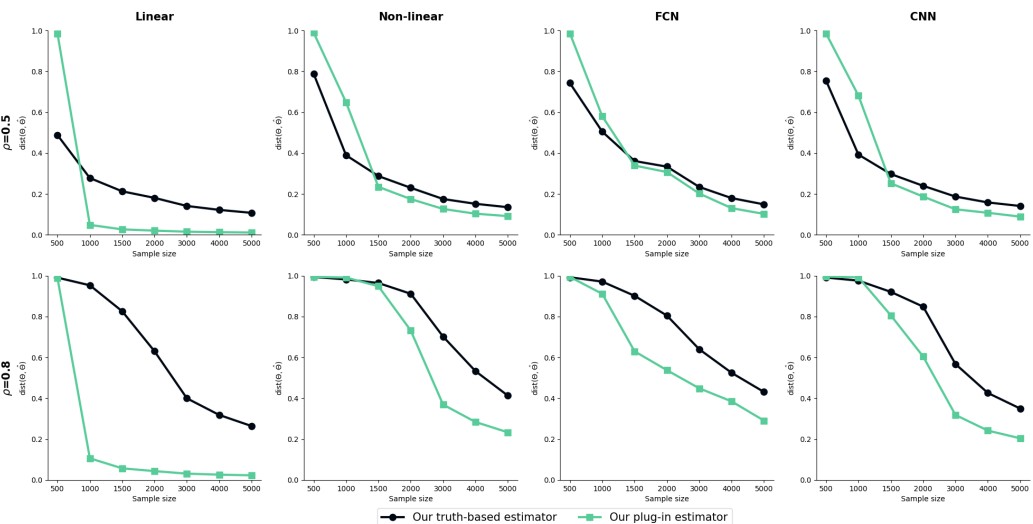

Figure 2: The original estimator vs the plug-in estimator under different covariance matrices.

By Figure 2, we are surprised to find that the plug-in estimator performs no worse, in fact even better, than the original estimator, particularly when the sample size $n$ increases. Indeed, as discussed in Section 4, in the Gaussian scenario, the plug-in estimator can be advantageous by offering reduced bias in linear models. This simulation further demonstrates that these advantages could extend beyond linear models, highlighting the adaptability of our approach when dealing with unknown parameters.

## 6 THE ADNI DATA ANALYSIS

In this section, we analyze brain imaging data from the Alzheimer's Disease Neuroimaging Initiative (ADNI) to demonstrate the performance of the proposed estimators. The ADNI[1] is a study aimed at identifying and monitoring Alzheimer's disease through clinical assessments, genetic analysis, imaging data, and more. Besides, another RetinaMNIST data analysis is provided in Section C.4.

In our analysis, we focus on using brain imaging data, magnetic resonance imaging (MRI), to predict the progression of Alzheimer's disease. After preprocessing T1-weighted MRI scans, we obtained 10,590 2D MRI images of size $48 \times 48$. Details of the preprocessing steps can be found in Section C.3. We incorporate real Mini-Mental State Examination (MMSE) scores as responses, ranging in $[0, 30]$. In this scenario, the estimation performance cannot be measured since the latent convolution filters are unknown. Thus we assess the prediction performance of our approach under various model configurations. Specifically, we predict the outcome $y_i$ using $G(\boldsymbol{X}^{\mathrm{ori}}) = f(\boldsymbol{X}^{\mathrm{ori}} \star \boldsymbol{B}_1, \ldots, \boldsymbol{X}^{\mathrm{ori}} \star \boldsymbol{B}_R)$ with varying numbers of filters and different link functions $f$. We consider the link function $f$ taking the forms of: linear model (I), the two-layer fully connected network (III), and the convolutional neural network (IV) discussed before. We use the proposed approach to first learn convolutional representations, which are then taken as inputs to these models for final prediction.

---

[1]https://adni.loni.usc.edu/

Moreover, we also predict the outcome $y_i$ using the original input without the convolution step, i.e., $G(\boldsymbol{X}^{\mathrm{ori}}) = \tilde{f}(\boldsymbol{X}^{\mathrm{ori}})$. Although the input dimensions differ with or without the convolution step, we maintain the same structure for the link functions $\tilde{f}$. In other words, we let $\tilde{f}$ take the same form of linear model, the two-layer FCN, and CNN, respectively. We note that the proposed approach achieves significant dimension reduction. For example, when fixing the convolution filters $\boldsymbol{B}_r$ to have a size of $4 \times 4$, the convolution operation allows to reduce the input dimension from 2,304 $(= 48^2)$ to $144 \times R$ for $R \in \{1, 2, 3\}$. We calculate the root mean square error (RMSE) of prediction for different approaches on the test set, obtained through 10-fold cross-validation. The average and standard deviation of RMSE under various settings are reported in Table 1 below.

Table 1: Prediction errors of various models across different levels of dimension reduction.

|           | Linear        | FCN           | CNN           |
| --------- | ------------- | ------------- | ------------- |
| Original  | 3.624 (0.252) | 3.735 (0.527) | 3.888 (0.365) |
| $R = 1$   | 3.394 (0.245) | 3.672 (0.337) | 3.471 (0.323) |
| $R = 2$   | 3.300 (0.250) | 3.641 (0.297) | 3.425 (0.361) |
| $R = 3$   | 3.318 (0.256) | 3.536 (0.329) | 3.443 (0.260) |

Table 1 shows that our convolutional representations consistently improve the baseline performance across different models. Specifically, in the linear and CNN settings, the model with two filters achieves the lowest prediction error, whereas in the FCN setting, the model with three filters performs best. Overall, this real data analysis demonstrates that the proposed convolutional representation learning approach yields improved predictive accuracy, validating its effectiveness as a dimension reduction method for medical imaging data.

## 7    CONCLUSION AND DISCUSSION

In this paper, we introduce an efficient, training-free approach to learn convolutional representations through a one-step SVD. We theoretically demonstrate that our estimators achieve an optimal convergence rate, and also highlight the superior performance of our approach empirically.

Although our analysis concentrated on non-overlapping convolutions, it is also possible to extend our approach to the overlapping case. Let $*$ represent a general convolution operator that allows for possible overlaps, with the stride denoted as $s_1 \times s_2$. Let $\boldsymbol{B}_1, ..., \boldsymbol{B}_R \in \mathbb{R}^{d_1 \times d_2}$ be the $R$ convolution filters in the first layer of a CNN. Assume that $s_1, s_2$ are factors of $P_1 - d_1, P_2 - d_2$ respectively. For any matrix $\boldsymbol{M} \in \mathbb{R}^{P_1 \times P_2}$ and a filter $\boldsymbol{B} \in \mathbb{R}^{d_1 \times d_2}$, the overlapping convolution is defined as

$$\boldsymbol{M} * \boldsymbol{B} \in \mathbb{R}^{\tilde{p}_1 \times \tilde{p}_2}, \;\; (\boldsymbol{M} * \boldsymbol{B})_{j,k} = \left\langle \tilde{\boldsymbol{M}}_{j,k}^{d_1, d_2, s_1, s_2}, \boldsymbol{B} \right\rangle,$$

where $\tilde{p}_1 = (P_1 - d_1)/s_1 + 1$, $\tilde{p}_2 = (P_2 - d_2)/s_2 + 1$, and $\tilde{\boldsymbol{M}}_{j,k}^{d_1, d_2, s_1, s_2}$ is the $(j, k)$-th covered features with size $d_1 \times d_2$ for $j \in [\tilde{p}_1]$ and $k \in [\tilde{p}_2]$. Similar to equation (4), we consider a reshaping operator $\tilde{\mathcal{R}}_{(d_1, d_2, s_1, s_2)} : \mathbb{R}^{P_1 \times P_2} \to \mathbb{R}^{(\tilde{p}_1 \tilde{p}_2) \times (d_1 d_2)}$, defined as

$$\tilde{\mathcal{R}}_{(d_1, d_2, s_1, s_2)}(\boldsymbol{M}) = \left[ \mathrm{vec}\left( \tilde{\boldsymbol{M}}_{1,1}^{d_1, d_2, s_1, s_2} \right), \ldots, \mathrm{vec}\left( \tilde{\boldsymbol{M}}_{p_1, p_2}^{d_1, d_2, s_1, s_2} \right) \right]^{\top}.$$

Denote $\tilde{\boldsymbol{X}} = \tilde{\mathcal{R}}_{(d_1, d_2, s_1, s_2)}(\boldsymbol{X}^{\mathrm{ori}})$ and $\boldsymbol{b}_r = \mathrm{vec}(\boldsymbol{B}_r) \in \mathbb{R}^{d_1 d_2}$, the CNN output can be written as

$$G(\boldsymbol{X}^{\mathrm{ori}}) = f(\boldsymbol{X}^{\mathrm{ori}} * \boldsymbol{B}_1, \ldots, \boldsymbol{X}^{\mathrm{ori}} * \boldsymbol{B}_R) = f(\tilde{\boldsymbol{X}} \boldsymbol{b}_1, ..., \tilde{\boldsymbol{X}} \boldsymbol{b}_R).$$

If we assume the model $\mathbb{E}(Y) = f(\tilde{\boldsymbol{X}} \boldsymbol{b}_1, ..., \tilde{\boldsymbol{X}} \boldsymbol{b}_R)$, the Stein's formula could still be applied to estimate the column space of $(\boldsymbol{b}_1, \cdots, \boldsymbol{b}_R)$, provided that the score function $S(\widetilde{\boldsymbol{X}})$ is available. For instance, when the original input is Gaussian, the covariance of the reshaped matrix $\widetilde{\boldsymbol{X}}$ needs to be estimated, paying particular attention to the overlaps within $\widetilde{\boldsymbol{X}}$. This scenario presents a promising direction for our future research.

**Ethics statement:** This work is purely methodological and theoretical, and therefore does not involve ethical concerns. The medical imaging data used in the real data analysis are publicly available.

**Reproducibility statement:** In addition to the theory part presented in Section 3, further theoretical results are provided in Section B. The complete proof of all theorems, along with technical lemmas are provided in Section D. Besides the descriptions of experiments in Sections 5 and 6, more experimental results and details including network configurations, data preprocessing steps are provided in Section C. The code is included in the supplementary material with the submission.

**The use of Large Language Models (LLMs) statement:** LLMs were used solely to polish the writing of this manuscript.

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

## A    ADDITIONAL RELATED WORKS

**SVD-based methods in CNN research.** There is a line of existing work utilizing Singular value decomposition (SVD) in CNN studies from multiple perspectives. Denton et al. (2014); Zhang et al. (2015); Lin et al. (2018) applied SVD for model compression and acceleration, decomposing weight matrices into low-rank approximations to reduce the number of parameters, which helps reduce storage requirements and improve inference speed. Han et al. (2015); Tai et al. (2015); Wang et al. (2019) proposed new model pruning methods. By analyzing the singular values, these pruning methods identify and remove less important components of the filters, thereby improving generalization and reducing overfitting. Raghu et al. (2017); Praggastis et al. (2022) used SVD to interpret the internal network structures, providing insights into the feature representations by examining the singular values and vectors. Boutsidis and Gallopoulos (2008); Kajo et al. (2018) demonstrated that SVD can be utilized to initialize weights with better-conditioned matrices, potentially leading to faster convergence and improved training stability. Jere et al. (2020); Song et al. (2022) focused on analyzing robustness and generalization, showing that the distribution of singular values can assess the sensitivity of the network to input perturbations, thus aiding in the study of model robustness.

While existing works primarily focus on using SVD as a tool to optimize, interpret, and improve CNN performance, which often applies SVD directly to learned filters, our work differs. Instead, our goal is to estimate the CNN filters through SVD applied to the score function of the data. To our knowledge, this is the first approach to estimate CNN filters without relying on gradient descent.

# B  THE TRUNCATED FORM FOR HEAVY-TAILED DATA

For heavy-tailed data, the direct sample version $(1/n)\sum_{i=1}^{n} Y_i S(\boldsymbol{X}_i)$ may not be optimal for estimating $\mathbb{E}[YS(\boldsymbol{X})]$ due to a lack of concentration. In fact, not only can the input $\boldsymbol{X}$ be heavy-tailed, but the outcome $Y$ or the score function $S(\boldsymbol{X})$ may also exhibit heavy-tailed behavior due to the unknown link function $f$. These issues have been discussed in the literature, where the truncation argument has been proposed, as seen in works such as Catoni (2012); Minsker (2018); Yang et al. (2017a). We adopt similar strategies and propose an improved estimator of the truncated form.

We start by considering a class of non-decreasing functions $\phi : \mathbb{R} \to \mathbb{R}$ that satisfies

$$-\log(1 - x + x^2/2) \leqslant \phi(x) \leqslant \log(1 + x + x^2/2), \quad x \in \mathbb{R}.$$

A straightforward option that satisfies these inequalities is

$$\phi(x) = \begin{cases} -\log(1 - x + x^2/2), & x < 0, \\ \log(1 + x + x^2/2), & x \geqslant 0. \end{cases} \tag{14}$$

Now we introduce a linear mapping $\psi : \mathbb{R}^{p \times d} \to \mathbb{R}^{p \times d}$ based on the non-decreasing function $\phi$. For a given matrix $\boldsymbol{X} \in \mathbb{R}^{p \times d}$, consider the spectral decomposition of its Hermitian dilation:

$$\boldsymbol{X}^* = \begin{bmatrix} 0 & \boldsymbol{X} \\ \boldsymbol{X}^\top & 0 \end{bmatrix} = \boldsymbol{U}\boldsymbol{\Sigma}\boldsymbol{U}^\top.$$

Further, define $\tilde{\boldsymbol{X}} = \boldsymbol{U}(\phi \circ \boldsymbol{\Sigma})\boldsymbol{U}^\top$ and write it into block form as

$$\tilde{\boldsymbol{X}} = \boldsymbol{U}(\phi \circ \boldsymbol{\Sigma})\boldsymbol{U}^\top = \begin{bmatrix} \underbrace{\tilde{\boldsymbol{X}}_{11}}_{p \times p} & \underbrace{\tilde{\boldsymbol{X}}_{12}}_{p \times d} \\ \underbrace{\tilde{\boldsymbol{X}}_{21}}_{d \times p} & \underbrace{\tilde{\boldsymbol{X}}_{22}}_{d \times d} \end{bmatrix},$$

where $\phi \circ \boldsymbol{\Sigma}$ indicates that the non-decreaing function $\phi$ is applied element-wise on $\boldsymbol{\Sigma}$. Then we define $\psi(\boldsymbol{X}) = \widetilde{\boldsymbol{X}}_{12}$. Based on $\psi$, we introduce a new estimator $\tilde{\boldsymbol{\Theta}}$ for $\boldsymbol{\Theta}$ with a truncated form

$$\tilde{\boldsymbol{\Theta}} = \text{SVD}_{v,R}\left(\frac{1}{n}\sum_{i=1}^{n} \tau(Y_i S(\boldsymbol{X}_i))\right), \tag{15}$$

where

$$\tau(Y_i S(\boldsymbol{X}_i)) = 1/\theta \cdot \psi[\theta Y_i S(\boldsymbol{X}_i)], \tag{16}$$

and $\theta > 0$ is a thresholding parameter. In later analysis, we will provide theoretical analyses of $\tilde{\boldsymbol{\Theta}}$ and demonstrate its advantages over $\hat{\boldsymbol{\Theta}}$ in (9) under heavy-tailed distributions.

## B.1  SINGLE FILTER ESTIMATION WITHOUT SUB-GAUSSIAN ASSUMPTIONS

In Section 3.1, we provide theoretical guarantees for the original estimator under the sub-Gaussian score assumption. In this subsection, we focus on the truncated estimator, for which we can provide sharp error bounds even when both the outcomes and the scores exhibit heavy-tailed behavior. Denote the truncated version of the estimated filter as $\tilde{\boldsymbol{b}} = \tilde{\boldsymbol{\Theta}}$. In other words,

$$\tilde{\boldsymbol{b}} = \text{SVD}_{v,1}\left(\frac{1}{n}\sum_{i=1}^{n} \tau(Y_i S(\boldsymbol{X}_i))\right).$$

**Assumption B.1.** *Suppose the model $\mathbb{E}(Y_i) = f(\boldsymbol{X}_i \boldsymbol{b})$ for $i \in [n]$. Assume that there exists an absolute constant $M > 0$ such that $\mathbb{E}(Y_i^4) \leqslant M$ and $\mathbb{E}(S_{jk}^4(\boldsymbol{X}_i)) \leqslant M$ for all $i \in [n]$, $j \in [p]$ and $k \in [d]$.*

Assumption B.1 relaxed the sub-Gaussian assumption in Section 3.1 to a bounded fourth-moment assumption on the outcome $Y$ and the score $S(\boldsymbol{X})$. Under this assumption, heavy-tailed inputs could be included. We note that similar assumptions have been studied in the literature, such as Yang et al. (2017a); Fan et al. (2018; 2021; 2023). With the bounded moment assumption, we can present the theoretical results for the truncated estimator.

**Theorem B.2.** *Suppose the model* $\mathbb{E}(Y_i) = f(\boldsymbol{X}_i \boldsymbol{b})$ *for* $i \in [n]$*. Suppose that Assumption 3.1(c) holds. Further assume Assumption B.1 with constant $M$. Let $\theta = \sqrt{2 \log(2(p+d)/\delta)/(nMpd)}$. Then, with probability at least $1 - \delta$, we have*

$$\|\tilde{\boldsymbol{b}} - \boldsymbol{b}\|_2 = \mathcal{O}\left(\sqrt{\frac{d}{n} \cdot \log \frac{p+d}{\delta}}\right). \tag{17}$$

By Theorem B.2, the truncated estimator could achieve a convergence rate of $\mathcal{O}\left(\sqrt{d \log(p+d)/n}\right)$ with a carefully chosen truncation parameter $\theta$. In other words, even if both the outcome $Y_i$ and the score $S(\boldsymbol{X}_i)$ are heavy-tailed, a nearly optimal convergence rate could still be guaranteed under a bounded moment condition. On the other hand, it is worth noting that the bounded expectation assumption on $f$, referred to as Assumption 3.1(b), is not necessary in Theorem B.2. In other words, the convergence can be guaranteed solely under the constant derivatives Assumption 3.1(c) and the bounded scores Assumption B.1.

### B.2 MULTIPLE FILTERS ESTIMATION WITHOUT SUB-GAUSSIAN ASSUMPTIONS

In this subsection, we show that sharp theoretical error bounds derived in Section B.1 could be generalized to the multiple filters case. Recall that the truncated estimator is defined as $\tilde{\boldsymbol{\Theta}} = \mathrm{SVD}_{v,R}\left(\frac{1}{n}\sum_{i=1}^{n} \tau(Y_i S(\boldsymbol{X}_i))\right)$. Based on the bounded moment Assumption B.1 and SVD-type Assumption 3.3, we have the following theorem.

**Theorem B.3.** *Suppose the model* $\mathbb{E}(Y_i) = f(\boldsymbol{X}_i \boldsymbol{\Theta})$ *for* $i \in [n]$*. Suppose that Assumption 3.3 holds. Further assume Assumption B.1 with constant $M$. Let $\theta = \sqrt{2 \log(2(p+d)/\delta)/(nMpd)}$. Then, with probability at least $1 - \delta$, we have*

$$\inf_{\boldsymbol{H} \in \mathbb{H}_R} \|\tilde{\boldsymbol{\Theta}} \boldsymbol{H} - \boldsymbol{\Theta}\|_F = \mathcal{O}\left(\sqrt{\frac{Rd}{n} \cdot \log \frac{p+d}{\delta}}\right).$$

Theorem B.3 characterizes the convergence rate of the column space of $\tilde{\boldsymbol{\Theta}}$ for multiple filters under the general setting. Compared to Theorem 3.4, Theorem B.3 further demonstrates that a sharp convergence rate can be guaranteed even when both the outcomes and the scores exhibit heavy-tailed behavior.

## C MORE EXPERIMENTAL RESULTS AND DETAILS

### C.1 SIMULATION

#### C.1.1 NETWORK STRUCTURES

We show the specific structure of the neural networks used in simulation for generation and fitting respectively. Note that their matching counterparts for fitting are the same so we omit them but show mis-matching ones. The FCN consists of ReLU function first, followed by a hidden layer with half the number of nodes in the input layer, batch normalization layer, ReLU function and an output layer. The ReLU function in the front is used as the activation function for the input which is from convolution. We consider the mis-matching network where the difference is that the number of hidden nodes equals to the input layer. The CNN consists of a convolution layer, batch normalization layer, ReLU function and max-pooling layer, followed by two fully-connected layers of which the first one reduces the dimension by half. For the mis-matching network, the difference is that the first fully-connected layer keeps the dimension rather than halving. The structures of networks used are shown in Figure 3.

Besides, in all experiments, our methods were run on Intel Xeon Gold 5218R CPUs and Adam was run on NVIDIA GeForce RTX 3090 GPUs.

#### C.1.2 NON-GAUSSIAN CASE

Beyond the Gaussian case, we consider the input with heavier-tailed distributions, such as the Student's $t$ and Gamma distributions below.

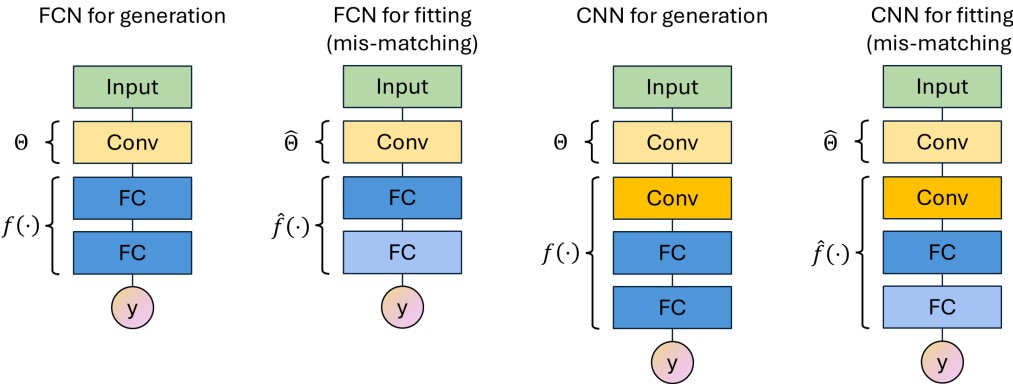

Figure 3: Structures of networks used.

(1) $\boldsymbol{X}^{\text{ori}}$ with i.i.d. Student's $t$ entries: $\boldsymbol{X}_{jk}^{\text{ori}} \sim t(\nu), \nu = 5$.

(2) $\boldsymbol{X}^{\text{ori}}$ with i.i.d. Gamma entries: $\boldsymbol{X}_{jk}^{\text{ori}} \sim \Gamma(\alpha, \beta)$ with $\alpha = 5$ and $\beta = 1$.

It is worth noting that while both t and Gamma distributions can display heavy-tailed behavior, the behavior of their scores differs. The score of a t distribution is bounded and inherently sub-Gaussian, whereas the score of a Gamma distribution remains heavy-tailed. Figure 4 below illustrates the estimation performance of our method and Adam for multiple-filter case.

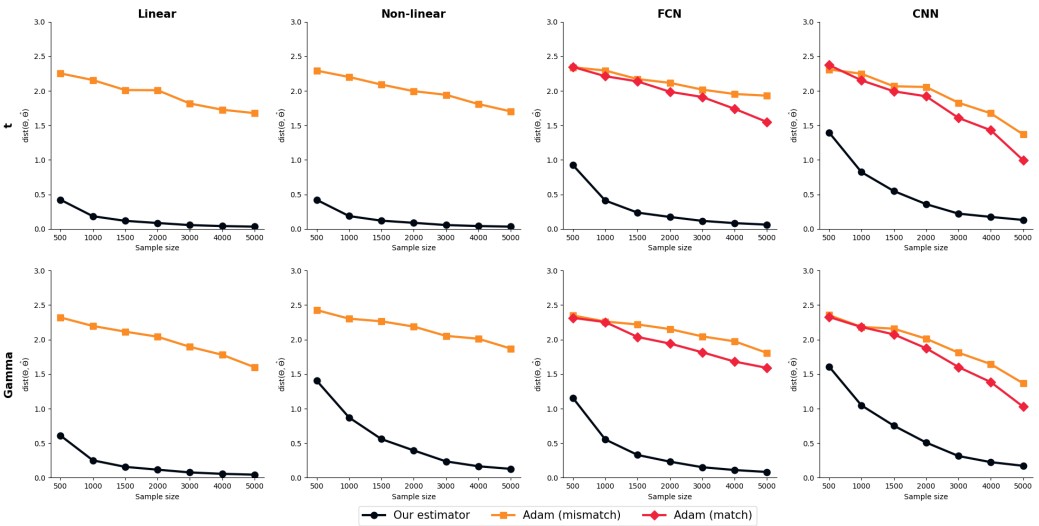

Figure 4: Convolution filter estimation performance of our approach and Adam under different link functions and non-Gaussian distributions.

### C.1.3 EFFECT OF TRUNCATION

As discussed earlier, when the data explicit heavy-tailed behavior, the original estimator may not be optimal due to a lack of concentration on the expectation of $YS(\boldsymbol{X})$. We compare the performance of the original and the truncated version of the estimator under different distribution assumptions. For the truncated estimator, we apply the non-decreasing function $\phi(\cdot)$ in (14). To be concise, simulations focus on the single filter case in this study, namely $y_i = f(\boldsymbol{X}_i^{\text{ori}} \star \boldsymbol{B}) + \varepsilon_i$.

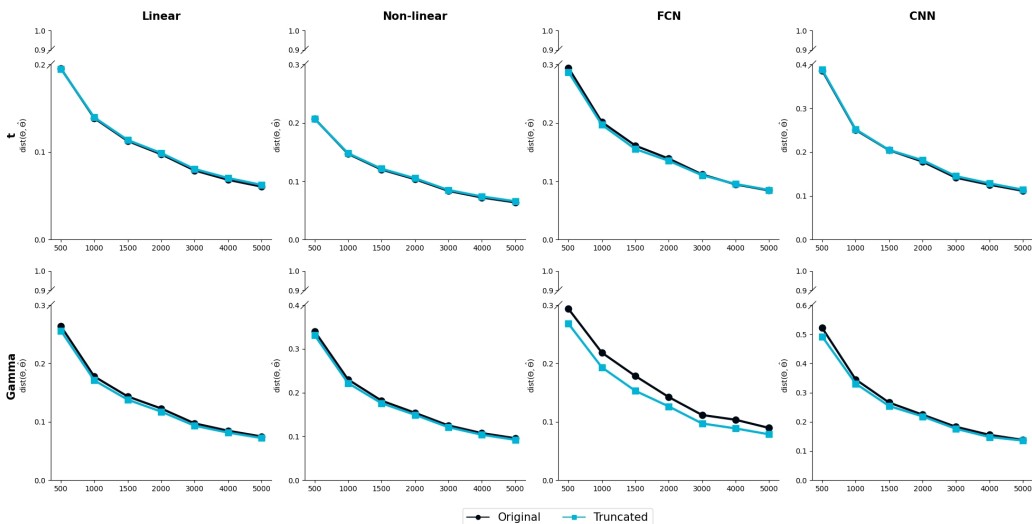

Figure 5: The original estimator vs the truncated estimator under the t and Gamma distribution with different link functions.

In Figure 5, we plot the estimation error of both the original and truncated estimators. We can see that the effect of truncation is relatively limited in the case of the t distribution, as the original and truncated estimators achieve similar estimation errors. In contrast, the truncated estimator generally achieves smaller estimation errors for the Gamma distribution case, particularly in the FCN scenario. This suggests that truncation is necessary when dealing with heavy-tailed scores.

## C.2 COMPUTATIONAL TIME COMPARISON

Since the computational times of each method show similar patterns across different combinations of link functions and distributions, we present results from a representative experiment under the first setting described in Section 5, where the data follow a correlated Gaussian distribution with a linear link function. While here we assume the covariance matrix to be unknown and must be estimated as part of our method. The Table C.2 below reports the average computational time (in seconds) over 100 repetitions. Our method consistently requires less than one second, substantially faster than Adam, which highlights its efficiency.

Table 2: Computational time comparison (in seconds)

| Sample size | 500 | 1000 | 1500 | 2000 | 3000 | 4000 | 5000 |
|---|---|---|---|---|---|---|---|
| Ours | 0.15 | 0.20 | 0.22 | 0.25 | 0.31 | 0.36 | 0.39 |
| Adam | 1.41 | 3.49 | 5.20 | 6.91 | 10.04 | 12.41 | 14.58 |

## C.3 ADNI DATASET

### C.3.1 PREPROCESSING

Before analysis, the brain MRI scans are carefully preprocessed following a standard pipeline involving denoising, resampling, bias-correction, affine-registration and unified segmentation, skull-stripping and cerebellum removing. It leads to 1059 images of size $48 \times 60 \times 48$ from normal subjects, ones with mild cognitive or Alzheimer's Disease. For each subject, we extract 10 middle coronal slices, which can be viewed as data augmentation of one slice per person. Therefore, we obtain the dataset consisting of 10590 samples, based on which we use 10-fold cross validation to compare different methods.

As for implementation, we mentioned that for simulated response we used the same networks as the simulation for generation and considered both matching and mis-matching networks for estimation. Also, we use the same networks as the simulation for generation to fit the real response. But note that without a known data model there is no so-called matching case.

### C.3.2 ADNI STUDY USING SIMULATED RESPONSE

Besides real MMSE response studied in Section 6, we also consider MRI images as input $\boldsymbol{X}_i^{\mathrm{ori}}$ and generate the outcomes $y_i$ according to model (13) with the link functions $f$ from (I) to (IV) in Section 5. As before, filters $(\boldsymbol{B}_1, \boldsymbol{B}_2, \boldsymbol{B}_3)$ are set to be of size $4 \times 4$ and consequently the reshaped images $\boldsymbol{X}_i = \mathcal{R}_{(4,4)}(\boldsymbol{X}^{\mathrm{ori}}) \in \mathbb{R}^{144 \times 16}$.

To validate our approach in real data analysis, a key step involves understanding the distributions of the input. We follow the work of Lindquist (2008) and assume that $\boldsymbol{X}_i^{\mathrm{ori}}$ follows a Gaussian distribution with unknown mean and covariance. We implement the plug-in estimator of our approach and compare its performance with Adam. Similarly, when the link functions are generated as fully-connected or convolutional neural networks, we implement Adam using both matched and non-matched structures for comparison. Table 3 illustrates the filter estimation performance under different models and link functions.

Table 3: (ADNI data analysis with simulated response) Convolution filter estimation performance of our plug-in estimator compared with Adam under different link functions.

|  | Linear | Non-linear | FCN | CNN |
|---|---|---|---|---|
| Proposed | 0.340 (0.018) | 0.387 (0.020) | 2.142 (0.155) | 0.716 (0.180) |
| Adam | 2.108 (0.145) | 2.137 (0.229) | 2.209 (0.207) | 2.189 (0.262) |
| Adam (match) | - | - | 2.348 (0.214) | 2.169 (0.188) |

As shown in Table 3, our approach attains the smallest estimation errors across various link functions. Despite the inputs being real MRI images with unknown distributions, our method continues to exhibit the best filter estimation performance, further highlighting its practical value.

### C.4 THE RETINAMNIST DATA ANALYSIS

To further demonstrate the effectiveness of our approach on medical images, we conducted an additional experiment using the RetinaMNIST dataset from MedMNIST[2], a widely used biomedical image collection formatted similarly to MNIST. This preprocessing centers and aligns the images, ensuring consistent size and structure across samples, and resizes them to a resolution of $28 \times 28$. The RetinaMNIST dataset consists of 1,600 retinal images labeled from 1 to 5, indicating the severity of diabetic retinopathy, with 1,080 samples used for training, 120 for validation, and 400 for testing.

We follow the same comparison framework as in the ADNI data analysis. We take this task as a regression problem and report the prediction RMSE on the test set in Table 4 below. The results show that our approach achieves improved performance consistently, highlighting its overall effectiveness.

Table 4: Prediction errors of various models across different levels of dimension reduction.

|  | Linear | FCN | CNN |
|---|---|---|---|
| Original | 1.533 | 1.693 | 1.480 |
| R=1 | 1.227 | 1.328 | 1.254 |
| R=2 | 1.191 | 1.401 | 1.186 |
| R=3 | 1.208 | 1.479 | 1.185 |

---

[2]https://medmnist.com/

# D  ADDITIONAL THEOREMS AND PROOFS

In this section, we first present the theoretical guarantee for the plug-in estimator under the case that Gaussian distribution with unknown covariances. Then we provide the proofs of the theorems and lemmas in Section 2, Section 3, Section B and Section D.1.

## D.1  THEORETICAL GUARANTEE FOR GAUSSIAN DISTRIBUTION WITH UNKNOWN COVARIANCES

To study the theoretical properties of $\breve{\boldsymbol{\Theta}}$, we impose the following condition on the link function $f(\cdot)$.

**Assumption D.1.** *Suppose the first-order Taylor expansion of the link function exist and denote as $f(\boldsymbol{Z}) = f(\boldsymbol{0}_{p \times R}) + \langle \nabla f(\boldsymbol{0}_{p \times R}), \boldsymbol{Z} \rangle + h(\boldsymbol{Z}) \| \boldsymbol{Z} \|_2^2$ with $\lim_{\| \boldsymbol{Z} \|_2 \to \boldsymbol{0}} h(\boldsymbol{Z}) = 0$. Assume that the remainder term is bounded, $h(\boldsymbol{X}_i \boldsymbol{\Theta}) \leqslant H$ for $i \in [n]$, where $H$ is a certain constant.*

**Theorem D.2.** *Suppose the model $\mathbb{E}(Y_i) = f(\boldsymbol{X}_i \boldsymbol{\Theta})$ for $i \in [n]$. Assume that $\boldsymbol{x}_i = vec(\boldsymbol{X}_i) \sim \mathcal{N}(\boldsymbol{0}, \boldsymbol{\Sigma})$ with $\lambda_{\min}(\boldsymbol{\Sigma}) > 0$. Let $\widehat{\boldsymbol{\Theta}}$ be the original estimator (9). Then under Assumption D.1, we have the following holds with probability approaching 1,*

$$\inf_{\boldsymbol{H} \in \mathbb{H}_R} \| \breve{\boldsymbol{\Theta}} \boldsymbol{H} - \widehat{\boldsymbol{\Theta}} \|_F = \mathcal{O} \left( \sqrt{\frac{Rd}{n}} \right). \tag{18}$$

## D.2  PROOF OF LEMMA 2.2

In order to prove Lemma 2.2, we first present the following lemma as an extension of Non-Gaussian Stein's Lemma (Stein et al., 2004). This version applies Stein's Lemma to non-Gaussian random matrices.

**Lemma D.3.** *Let $g : \mathbb{R}^{p \times d} \to \mathbb{R}$ be a continuously differentiable function and $\boldsymbol{X} \in \mathbb{R}^{p \times d}$ be a random matrix with density $P$ which is also continuously differentiable. Under the assumption that $\mathbb{E}[g(\boldsymbol{X}) \cdot S_{ij}(x)]$ and $\mathbb{E}[\nabla_{X_{ij}} g(\boldsymbol{X})]$ are well-defined, we have*

$$\mathbb{E}[g(\boldsymbol{X}) \cdot S_{ij}(\boldsymbol{X})] = \mathbb{E}[\nabla_{X_{ij}} g(\boldsymbol{X})].$$

***Proof of Lemma D.3.*** By the definition of $S_{ij}$, we know $S_{ij}(\boldsymbol{X}) = -\nabla_{X_{ij}} P(\boldsymbol{X}) / P(\boldsymbol{X})$. Thus, we have that

$$\mathbb{E}[g(\boldsymbol{X}) \cdot S_{ij}(\boldsymbol{X})] = \int_{\mathbb{R}^{p \times d}} g(x) S_{ij}(x) P(x) dx = -\int_{\mathbb{R}^{p \times d}} g(x) \nabla_{x_{ij}} P(x) dx.$$

By the integration by parts formula, we get

$$\int_{\mathbb{R}^{p \times d}} g(x) \nabla_{x_{ij}} P(x) dx = \int_{\mathbb{R}^{p \times d}} \nabla_{x_{ij}} \big[ g(x) P(x) \big] dx - \int_{\mathbb{R}^{p \times d}} \nabla_{x_{ij}} g(x) P(x) dx$$

$$= -\int_{\mathbb{R}^{p \times d}} \nabla_{x_{ij}} g(x) P(x) dx$$

$$= -\mathbb{E}[\nabla_{X_{ij}} g(\boldsymbol{X})].$$

Combining two equations, we can obtain that

$$\mathbb{E}[g(\boldsymbol{X}) \cdot S_{ij}(\boldsymbol{X})] = \mathbb{E}[\nabla_{X_{ij}} g(\boldsymbol{X})].$$

$\square$

Next, we are ready to prove Lemma 2.2.

***Proof of Lemma 2.2.*** By Lemma D.3, assuming that the link function $f : \boldsymbol{z} \mapsto f(\boldsymbol{z})$ and the score function $S(\boldsymbol{X})$ satisfy the conditions stated in Lemma D.3, we can get

$$\mathbb{E}[Y \cdot S_{ij}(\boldsymbol{X})] = \mathbb{E}[f(\boldsymbol{X}\boldsymbol{\Theta}) \cdot S_{ij}(\boldsymbol{X})] = \mathbb{E}\{\nabla_{X_{ij}} [f(\boldsymbol{X}\boldsymbol{\Theta})]\} = \mathbb{E} \left[ \frac{\partial f}{\partial \boldsymbol{z}_{i\cdot}} (\boldsymbol{X}\boldsymbol{\Theta}) \right] \cdot \boldsymbol{\Theta}_{j\cdot}^\top,$$

where $z_i.$ is the $i$-th row of $z$ and $\Theta_{j\cdot}$ is the $j$-th row of $\Theta$. Let $S_{\cdot j}(X)$ be the $j$-th column of $S(X)$. We can get that for all $j \in [d]$,

$$\mathbb{E}[Y \cdot S_{\cdot j}(X)] = \mathbb{E}\left[\nabla_z f(X\Theta)\right] \cdot \Theta_{j\cdot}^\top.$$

Then, combining all the vectors into a matrix, we obtain that

$$\mathbb{E}[Y \cdot S(X)] = \mathbb{E}\left[\nabla_z f(X\Theta)\right] \cdot \Theta^\top.$$

$\square$

### D.3 PROOF OF THEOREM 3.2

We first prove a lemma which gives a bound for the moments of Gaussian random variable.

**Lemma D.4.** *If $Z \sim N(0, \sigma^2)$, then $[\mathbb{E}|Z|^k]^{\frac{1}{k}} \leqslant \sigma\sqrt{k}$ for $k \in \mathbb{R}_+$.*

***Proof of Lemma D.4.*** We can get that

$$\mathbb{E}|Z|^k = \int_{\mathbb{R}} |z|^k \frac{1}{\sqrt{2\pi}\sigma} \exp\left(-\frac{z^2}{2\sigma^2}\right) dz$$

$$= 2 \int_0^\infty z^k \frac{1}{\sqrt{2\pi}\sigma} \exp\left(-\frac{z^2}{2\sigma^2}\right) dz$$

$$\overset{(i)}{=} 2 \int_0^\infty \frac{1}{\sqrt{2\pi}\sigma} (\sqrt{2t}\sigma)^k e^{-t} \cdot \frac{\sqrt{2}\sigma}{2\sqrt{t}} dt$$

$$= \frac{(\sqrt{2}\sigma)^k}{\sqrt{\pi}} \cdot \int_0^\infty t^{\frac{k-1}{2}} e^{-t} dt$$

$$\overset{(ii)}{=} \frac{(\sqrt{2}\sigma)^k}{\sqrt{\pi}} \cdot \Gamma\left(\frac{k+1}{2}\right),$$

where $(i)$ is by the transformation $z = \sqrt{2t}\sigma$, $(ii)$ is by $\Gamma(x) = \int_0^\infty t^{x-1} e^{-t} dt$.

If $k$ is even, then

$$\frac{(\sqrt{2}\sigma)^k}{\sqrt{\pi}} \cdot \Gamma\left(\frac{k+1}{2}\right) = \frac{(\sqrt{2}\sigma)^k}{\sqrt{\pi}} \cdot \frac{k-1}{2} \cdot \frac{k-3}{2} \cdots \frac{1}{2} \cdot \Gamma\left(\frac{1}{2}\right)$$

$$\leqslant \frac{(\sqrt{2}\sigma)^k}{\sqrt{\pi}} \cdot \frac{\sqrt{k!}}{2^{k/2}} \cdot \sqrt{\pi}$$

$$= \sigma^k \sqrt{k!}.$$

If $k$ is odd, then

$$\frac{(\sqrt{2}\sigma)^k}{\sqrt{\pi}} \cdot \Gamma\left(\frac{k+1}{2}\right) = \frac{(\sqrt{2}\sigma)^k}{\sqrt{\pi}} \cdot \frac{k-1}{2} \cdot \frac{k-3}{2} \cdots 1 \cdot \Gamma(1)$$

$$\leqslant \frac{(\sqrt{2}\sigma)^k}{\sqrt{\pi}} \cdot \frac{\sqrt{k!}}{2^{(k-1)/2}}$$

$$\leqslant \sigma^k \sqrt{k!}.$$

Thus,

$$[\mathbb{E}|Z|^k]^{\frac{1}{k}} \leqslant (\sigma^k \sqrt{k!})^{\frac{1}{k}} \leqslant \sigma\sqrt{k}.$$

$\square$

Then, we show the convergence rate of $\frac{1}{n}\sum_{i=1}^n Y_i S(X_i)$ by the following lemma.

**Lemma D.5.** *Suppose the model $\mathbb{E}(Y_i) = f(X_i\Theta)$ for $i \in [n]$. Under Assumption (a), Assumption (b) and Assumption (c), with probability at least $1 - \delta$, we have*

$$\left\|\frac{1}{n}\sum_{i=1}^n Y_i S(X_i) - \mathbb{E}[Y S(X)]\right\|_F = \mathcal{O}\left(\sqrt{\frac{pd}{n}} \cdot \log \frac{pd}{\delta}\right).$$

***Proof of Theorem D.5***. Since $S_{ij}(\boldsymbol{X})$ is a sub-Gaussian random variable, we can get that for any $i \in [p], j \in [d]$, there exists a positive constant $L$ such that $(\mathbb{E}|S_{ij}(\boldsymbol{X})|^k)^{\frac{1}{k}} \leqslant L\sqrt{k}$ for $k \geqslant 1$. We first prove that $YS_{ij}(\boldsymbol{X}) - \mathbb{E}[YS_{ij}(\boldsymbol{X})]$ is a sub-exponential random variable. To simplify the notation, in the following proof, we denote $f = f(\boldsymbol{X\Theta})$ and $s = S_{ij}(\boldsymbol{X})$. Then, for $k \geqslant 1$, we have that

$$
\begin{aligned}
\left[\mathbb{E}|Ys - \mathbb{E}(Ys)|^k\right]^{\frac{1}{k}} &\overset{(i)}{=} \left[\mathbb{E}|(f+\varepsilon)s - \mathbb{E}(fs)|^k\right]^{\frac{1}{k}} \\
&\overset{(ii)}{\leqslant} (\mathbb{E}|fs|^k)^{\frac{1}{k}} + (\mathbb{E}|\varepsilon s|^k)^{\frac{1}{k}} + |\mathbb{E}(fs)| \\
&\overset{(iii)}{\leqslant} (\mathbb{E}f^{2k} \cdot \mathbb{E}s^{2k})^{\frac{1}{2k}} + (\mathbb{E}|\varepsilon|^k \cdot \mathbb{E}|s|^k)^{\frac{1}{k}} + (\mathbb{E}f^2 \cdot \mathbb{E}s^2)^{\frac{1}{2}} \\
&\overset{(iv)}{\leqslant} T \cdot L\sqrt{2k} + \sigma_\varepsilon\sqrt{k} \cdot L\sqrt{k} + T \cdot L\sqrt{2} \\
&\overset{(v)}{\leqslant} 4TLk + \sigma_\varepsilon Lk \\
&= (4T + \sigma_\varepsilon)Lk,
\end{aligned}
$$

where $(i)$ is by $\mathbb{E}(\varepsilon s) = \mathbb{E}\varepsilon \cdot \mathbb{E}s = 0$, $(ii)$ is by Minkowski inequality, $(iii)$ is by Cauchy–Schwarz inequality and the independence of $\varepsilon$ and $\boldsymbol{X}$, $(iv)$ is by Assumption (a), Assumption (b) and Lemma D.4 , $(v)$ is by $k \geqslant 1$. Thus, $YS_{ij}(\boldsymbol{X}) - \mathbb{E}[YS_{ij}(\boldsymbol{X})]$ is a sub-exponential random variable and we can get

$$
\begin{aligned}
\|YS_{ij}(\boldsymbol{X}) - \mathbb{E}[YS_{ij}(\boldsymbol{X})]\|_{\psi_1} &= \sup_{k \geqslant 1} \frac{1}{k}\left(\mathbb{E}|YS_{ij}(\boldsymbol{X}) - \mathbb{E}[YS_{ij}(\boldsymbol{X})]|^k\right)^{\frac{1}{k}} \\
&\leqslant 4T + \sigma_\varepsilon.
\end{aligned}
$$

Then, by Bernstein's inequality, for any $t > 0$, we have

$$
\mathbb{P}\left(\left|\frac{1}{n}\sum_{k=1}^n Y_k S_{ij}(\boldsymbol{X}_k) - \mathbb{E}[YS_{ij}(\boldsymbol{X})]\right| \geqslant t\right) \leqslant 2 \cdot \exp\left[-cn\min\left(\frac{t^2}{K^2}, \frac{t}{K}\right)\right],
$$

where $K \leqslant 4T + \sigma_\varepsilon$ and $c$ is a positive constant. Therefore, we can obtain that

$$
\begin{aligned}
&\mathbb{P}\left(\left\|\frac{1}{n}\sum_{k=1}^n Y_k S(\boldsymbol{X}_k) - \mathbb{E}(YS(\boldsymbol{X}))\right\|_F \geqslant \sqrt{pd} \cdot t\right) \\
&\leqslant \sum_{i=1}^p \sum_{j=1}^d \mathbb{P}\left(\left|\frac{1}{n}\sum_{k=1}^n Y_k S_{ij}(\boldsymbol{X}_k) - \mathbb{E}[YS_{ij}(\boldsymbol{X})]\right| \geqslant t\right) \\
&\leqslant 2pd \cdot \exp\left[-cn\min\left(\frac{t^2}{K^2}, \frac{t}{K}\right)\right].
\end{aligned}
$$

Letting $t = \frac{4T+\sigma_\varepsilon}{\sqrt{cn}} \cdot \sqrt{\log \frac{2pd}{\delta}}$, we get

$$
\mathbb{P}\left(\left\|\frac{1}{n}\sum_{k=1}^n Y_k S(\boldsymbol{X}_k) - \mathbb{E}(YS(\boldsymbol{X}))\right\|_F \geqslant \sqrt{\frac{pd}{cn}}(4T+\sigma_\varepsilon)\sqrt{\log\frac{2pd}{\delta}}\right) \leqslant \delta.
$$

Therefore, we can conclude that with probability at least $1 - \delta$,

$$
\left\|\frac{1}{n}\sum_{k=1}^n Y_k S(\boldsymbol{X}_k) - \mathbb{E}(YS(\boldsymbol{X}))\right\|_F = \mathcal{O}\left(\sqrt{\frac{pd}{n} \cdot \log\frac{pd}{\delta}}\right).
$$

$\square$

By Lemma D.5, we utilize a modified Davis-Kahan theorem proposed in O'Rourke et al. (2018) to prove Theorem 3.2.

**Proof of Theorem 3.2.** To simplify the notation, we denote $A = \mathbb{E}[YS(\boldsymbol{X})]$, $A_i = Y_i S(\boldsymbol{X}_i)$, and $\hat{A} = \frac{1}{n} \sum_{i=1}^{n} A_i$. Then, by Theorem 4 in O'Rourke et al. (2018), we can get

$$\sin \Theta(\widehat{\boldsymbol{b}}, \boldsymbol{b}) \leqslant 2 \cdot \frac{\|\hat{A} - A\|_{op}}{\|A\|_{op}}.$$

Since $\|\hat{A} - A\|_{op} \leqslant \|\hat{A} - A\|_F$ and $\|A\|_{op} \geqslant C_1 \sqrt{p}$, we can obtain that with probability $1 - \delta$,

$$\sin \Theta(\widehat{\boldsymbol{b}}, \boldsymbol{b}) \leqslant \frac{2}{C_1}(4T + \sigma_\varepsilon)\sqrt{\frac{d}{cn} \log \frac{2pd}{\delta}}.$$

We can assume that $\cos \Theta(\widehat{\boldsymbol{b}}, \boldsymbol{b}) = \langle \widehat{\boldsymbol{b}}, \boldsymbol{b} \rangle \geqslant 0$, otherwise we use $-\widehat{\boldsymbol{b}}$ to replace $\widehat{\boldsymbol{b}}$. Then,

$$\|\widehat{\boldsymbol{b}} - \boldsymbol{b}\|_2 = \sqrt{\|\widehat{\boldsymbol{b}}\|_2^2 + \|\boldsymbol{b}\|_2^2 - 2\langle \widehat{\boldsymbol{b}}, \boldsymbol{b} \rangle}$$

$$= \sqrt{2 - 2\cos \Theta(\widehat{\boldsymbol{b}}, \boldsymbol{b})}$$

$$\leqslant \sqrt{2} \sin \Theta(\widehat{\boldsymbol{b}}, \boldsymbol{b})$$

$$\leqslant \frac{2\sqrt{2}}{C_1}(4T + \sigma_\varepsilon)\sqrt{\frac{d}{cn} \log \frac{2pd}{\delta}},$$

Thus, we have $\|\widehat{\boldsymbol{b}} - \boldsymbol{b}\|_2 = \mathcal{O}(\sqrt{d/n \cdot \log(pd/\delta)})$. $\qquad \square$

### D.4 PROOF OF THEOREM B.2

We first get the following lemma to show the convergence rate of $\frac{1}{n} \sum_{i=1}^{n} \tau(Y_i S(\boldsymbol{X}_i))$.

**Lemma D.6.** *Suppose the model* $\mathbb{E}(Y_i) = f(\boldsymbol{X}_i \boldsymbol{\Theta})$ *for* $i \in [n]$. *Under Assumption B.1 and Assumption (c), with probability at least* $1 - \delta$, *letting*

$$\theta = \sqrt{\frac{2 \log(2(p + d)/\delta)}{nMpd}},$$

*we have*

$$\left\| \frac{1}{n} \sum_{i=1}^{n} \tau(Y_i S(\boldsymbol{X}_i)) - \mathbb{E}[YS(\boldsymbol{X})] \right\|_{op} = \mathcal{O}\left( \sqrt{\frac{pd}{n} \cdot \log \frac{p + d}{\delta}} \right).$$

**Proof of Theorem D.6.** For any $u \in \mathbb{R}^p$ such that $\|u\|_2 = 1$, we have that

$$u^\top \mathbb{E}[YS(\boldsymbol{X})(YS(\boldsymbol{X}))^\top]u = \mathbb{E}[Y^2 \cdot u^\top S(\boldsymbol{X})S(\boldsymbol{X})^\top u]$$

$$= \sum_{j=1}^{d} \mathbb{E}[Y^2 \cdot (S_{\cdot,j}(\boldsymbol{X})^\top u)^2]$$

$$\leqslant \sum_{j=1}^{d} \sqrt{\mathbb{E}(Y^4) \cdot \mathbb{E}[(S_{\cdot,j}(\boldsymbol{X})^\top u)^4]},$$

where the last inequality is by Cauchy–Schwarz inequality. And, for any $j \in [d]$,

$$\mathbb{E}[(S_{\cdot,j}(\boldsymbol{X})^\top u)^4] = \mathbb{E}\left[ \left( \sum_{i=1}^{p} S_{i,j}(\boldsymbol{X}) u_i \right)^4 \right]$$

$$\overset{(i)}{\leqslant} \mathbb{E}\left[ \left( \sum_{i=1}^{p} S_{i,j}^2(\boldsymbol{X}) \right)^2 \left( \sum_{i=1}^{p} u_i^2 \right)^2 \right]$$

$$\overset{(ii)}{\leqslant} \mathbb{E}\left[ p \sum_{i=1}^{p} S_{i,j}^4(\boldsymbol{X}) \right]$$

$$\overset{(iii)}{\leqslant} Mp^2,$$

where $(i)$ is by Cauchy-Schwarz inequality, $(ii)$ is by Cauchy-Schwarz inequality and $\|u\|_2 = 1$, $(iii)$ is by $\mathbb{E}S_{ij}^4(\boldsymbol{X}) \leqslant M$.

Thus, we obtain that $u^\top \mathbb{E}[YS(\boldsymbol{X})(YS(\boldsymbol{X}))^\top]u \leqslant Mpd$, which implies that $\|\mathbb{E}[YS(\boldsymbol{X})(YS(\boldsymbol{X}))^\top]\|_{op} \leqslant Mpd$. Similarly, we can get $\|\mathbb{E}[(YS(\boldsymbol{X}))^\top YS(\boldsymbol{X})]\|_{op} \leqslant Mpd$. Therefore, by Corollary 3.1 in Minsker (2018), we can get that

$$\mathbb{P}\left(\left\|\frac{1}{\theta n}\sum_{i=1}^n \psi[\theta Y_i S(\boldsymbol{X}_i)] - \mathbb{E}[YS(\boldsymbol{X})]\right\|_{op} \geqslant \frac{t}{\sqrt{n}}\right) \leqslant 2(p+d)\exp\left[-\theta t\sqrt{n} + \frac{\theta^2 nMpd}{2}\right]$$

for any $t > 0$ and $\theta > 0$. Setting $\theta = \sqrt{2\log(2(p+d)/\delta)}/\sqrt{nMpd}$ and $t = \sqrt{2Mpd\log(2(p+d)/\delta)}$, we have that

$$\mathbb{P}\left(\left\|\frac{1}{n}\sum_{i=1}^n \tau(Y_i S(\boldsymbol{X}_i)) - \mathbb{E}[YS(\boldsymbol{X})]\right\|_{op} \geqslant \sqrt{\frac{2Mpd}{n}\log\frac{2(p+d)}{\delta}}\right) \leqslant \delta,$$

which also means that with probability at least $1 - \delta$,

$$\left\|\frac{1}{n}\sum_{i=1}^n \tau(Y_i S(\boldsymbol{X}_i)) - \mathbb{E}[YS(\boldsymbol{X})]\right\|_{op} = \mathcal{O}\left(\sqrt{\frac{pd}{n}\cdot\log\frac{p+d}{\delta}}\right).$$

$\square$

Then, similar to the proof of Theorem 3.2, we can also prove Theorem B.2 by directly using the modified Davis-Kahan theorem.

***Proof of Theorem B.2***. The proof is the same as that of Theorem 3.2. We can obtain that

$$\|\tilde{\boldsymbol{b}} - \boldsymbol{b}\|_2 \leqslant \sqrt{2}\sin\Theta(\tilde{\boldsymbol{b}} - \boldsymbol{b})$$

$$\leqslant 2\sqrt{2}\cdot\frac{\|\frac{1}{n}\sum_{i=1}^n \tau(Y_i S(\boldsymbol{X}_i)) - \mathbb{E}[YS(\boldsymbol{X})]\|_{op}}{\|\mathbb{E}[YS(\boldsymbol{X})]\|_{op}}$$

$$= \mathcal{O}\left(\sqrt{\frac{d}{n}\cdot\log\frac{p+d}{\delta}}\right).$$

$\square$

## D.5 PROOF OF THEOREM 3.4

Based on Lemma D.5 and a modified Davis–Kahan–Wedin sine theorem proposed in O'Rourke et al. (2018), we can prove Theorem 3.4 as follows.

***Proof of Theorem 3.4***. To simplify the notation, we denote $A = \mathbb{E}[YS(\boldsymbol{X})]$, $A_i = Y_i S(\boldsymbol{X}_i)$, and $\hat{A} = \frac{1}{n}\sum_{i=1}^n A_i$. According to Lemma D.5, we have that with probability at least $1 - \delta$,

$$\|\hat{A} - A\|_F = \mathcal{O}\left(\sqrt{\frac{pd}{n}\cdot\log\frac{pd}{\delta}}\right).$$

Then, by Theorem 19 in O'Rourke et al. (2018), we can obtain that

$$\|\widehat{\boldsymbol{\Theta}}\hat{\boldsymbol{O}} - \boldsymbol{\Theta}\|_F \leqslant \frac{2^{3/2}\|\hat{A} - A\|_F}{\sigma_k} = \mathcal{O}\left(\sqrt{\frac{pd}{n}\cdot\log\frac{pd}{\delta}}\right).$$

$\square$

## D.6 Proof of Theorem B.3

Based on Lemma D.6 and a modified Davis–Kahan–Wedin sine theorem proposed in O'Rourke et al. (2018), we can prove Theorem B.3 as follows.

***Proof of Theorem B.3.*** To simplify the notation, we denote $A = \mathbb{E}[YS(\boldsymbol{X})]$ and $\tilde{A} = \frac{1}{n}\sum_{i=1}^{n}\tau(Y_i S(\boldsymbol{X}_i))$. According to Lemma D.6, we can get that with probability at least $1 - \delta$,

$$\|\tilde{A} - A\|_F = \mathcal{O}\left(\sqrt{\frac{pd}{n}\cdot\log\frac{p+d}{\delta}}\right).$$

Then, by Theorem 19 in O'Rourke et al. (2018), we can obtain that

$$\|\tilde{\boldsymbol{\Theta}}\tilde{\boldsymbol{O}} - \boldsymbol{\Theta}\|_F \leqslant \frac{2^{3/2}\|\tilde{A} - A\|_F}{\sigma_k} = \mathcal{O}\left(\sqrt{\frac{pd}{n}\cdot\log\frac{p+d}{\delta}}\right).$$

$\square$

## D.7 Proof of Theorem D.2

**Lemma D.7.** *(Vershynin (2020), Covariance estimation). Let $\boldsymbol{X}$ be an $n \times p$ matrix whose rows $\boldsymbol{x}_i$ are independent, mean zero, sub-Gaussian random vectors in $\mathbb{R}^p$ with the same covariance matrix $\boldsymbol{\Sigma}$. Denote $\widehat{\boldsymbol{\Sigma}} = (1/n)\boldsymbol{X}^\top\boldsymbol{X}$ as the sample covariance matrix. Then for any $t \geqslant 0$ we have*

$$\left\|\widehat{\boldsymbol{\Sigma}} - \boldsymbol{\Sigma}\right\|_{op} \leqslant \max(\delta, \delta^2), \quad where\ \delta = CLt\sqrt{\frac{p}{n}} \tag{19}$$

*with probability at least $1 - 2\exp(-t^2 p)$. Here $L = \max(K, K^2)$ and $K = \max_i \|\boldsymbol{x}_i\|_{\psi_2}$.*

**Lemma D.8.** *(Vershynin (2020)) Let $\boldsymbol{X}$ be an $n \times p$ matrix whose rows $\boldsymbol{x}_i$ are independent isotropic random vectors in $\mathbb{R}^p$. Assume that for some $L \geqslant 0$, it holds almost surely for every $i$ that $\|\boldsymbol{x}_i\|_2 \leqslant L\sqrt{p}$. Then for every $t \geqslant 0$, one has*

$$\sqrt{n} - tL\sqrt{p} \leqslant \sigma_{\min}(\boldsymbol{X}) \leqslant \sigma_{\max}(\boldsymbol{X}) \leqslant \sqrt{n} + tL\sqrt{p}$$

*with probability at least $1 - 2p \cdot \exp(-ct^2)$.*

**Lemma D.9.** *(Xu (2020), Perturbation of inverse) For two matrices $\boldsymbol{A}, \boldsymbol{B} \in \mathbb{R}^{p \times p}$, we have*

$$\left\|\boldsymbol{B}^{-1} - \boldsymbol{A}^{-1}\right\|_2 \leqslant \|\boldsymbol{A}\|_2 \|\boldsymbol{B}\|_2 \|\boldsymbol{B} - \boldsymbol{A}\|_2 \tag{20}$$

**Lemma D.10.** *Let $\boldsymbol{x} \in \mathbb{R}^p$ be a sub-Gaussian random vector with parameter $\sigma$, then with probability at least $1 - t$ for $t \in (0, 1)$*

$$\|\boldsymbol{x}\|_2 \leqslant 4\sigma\sqrt{p} + 2\sigma\sqrt{\log(1/t)} \tag{21}$$

***Proof of Theorem D.2.*** According to Taylor's theorem, $f(\boldsymbol{X}_i\boldsymbol{\Theta})$ can be expanded at the point $\boldsymbol{a} = \boldsymbol{0}_p$ as follows with a function $h : \mathbb{R}^{p \times R} \to \mathbb{R}$

$$f(\boldsymbol{X}_i\boldsymbol{\Theta}) = \langle\nabla f(\boldsymbol{0}_{p\times R}), \boldsymbol{X}_i\boldsymbol{\Theta}\rangle + h(\boldsymbol{X}_i\boldsymbol{\Theta})\|\boldsymbol{X}_i\boldsymbol{\Theta}\|$$

$$= \boldsymbol{x}_i^\top\sum_{r=1}^{R}\left(\boldsymbol{b}_r \otimes \nabla f(\boldsymbol{0}_{p\times R})_{[:,r]}\right) + h(\boldsymbol{X}_i\boldsymbol{\Theta})\|\boldsymbol{X}_i\boldsymbol{\Theta}\|,$$

with $\lim_{\boldsymbol{X}_i\boldsymbol{\Theta}\to\boldsymbol{0}} h(\boldsymbol{X}_i\boldsymbol{\Theta}) = 0$, where $\nabla f(\boldsymbol{0}_{p\times R})_{[:,r]}$ is $r$-th column of $\nabla f(\boldsymbol{0}_{p\times R})$. Let $\boldsymbol{Y} = (Y_i, \ldots, Y_n)^\top$, $\boldsymbol{X} = (\boldsymbol{x}_1, \ldots, \boldsymbol{x}_n)^\top$ and $\boldsymbol{R} = (h(\boldsymbol{X}_1\boldsymbol{\Theta})\|\boldsymbol{X}_1\boldsymbol{\Theta}\|, \ldots, h(\boldsymbol{X}_n\boldsymbol{\Theta})\|\boldsymbol{X}_n\boldsymbol{\Theta}\|)$, then we can write the model in a matrix form below

$$\boldsymbol{Y} = \boldsymbol{X}\sum_{r=1}^{R}\left(\boldsymbol{b}_r \otimes \nabla f(\boldsymbol{0}_{p\times R})_{[:,r]}\right) + \boldsymbol{R} + \boldsymbol{\varepsilon}.$$

Let $\boldsymbol{S} = (S(\boldsymbol{x}_1), \ldots, S(\boldsymbol{x}_n))^{\top}$ be the full score matrix, it holds for Gaussian distribution with zero mean and covariance $\boldsymbol{\Sigma}$ that $\boldsymbol{S} = \boldsymbol{X}\boldsymbol{\Sigma}^{-1}$. Therefore, we have the truth-based estimator by algebra

$$\text{vec}\left[\frac{1}{n}\sum_{i=1}^{n}Y_i S(\boldsymbol{X}_i)\right] = \boldsymbol{\Sigma}^{-1}\widehat{\boldsymbol{\Sigma}}\sum_{r=1}^{R}\left(\boldsymbol{b}_r \otimes \nabla f(\boldsymbol{0}_{p\times R})_{[:,r]}\right) + \frac{1}{n}\boldsymbol{\Sigma}^{-1}\boldsymbol{X}^{\top}(\boldsymbol{R}+\boldsymbol{\varepsilon}). \quad (22)$$

If we replace $\boldsymbol{\Sigma}$ by $\widehat{\boldsymbol{\Sigma}}$, we can directly get the plug-in estimator below

$$\text{vec}\left[\frac{1}{n}\sum_{i=1}^{n}Y_i \hat{S}(\boldsymbol{X}_i)\right] = \sum_{r=1}^{R}\left(\boldsymbol{b}_r \otimes \nabla f(\boldsymbol{0}_{p\times R})_{[:,r]}\right) + \frac{1}{n}\widehat{\boldsymbol{\Sigma}}^{-1}\boldsymbol{X}^{\top}(\boldsymbol{R}+\boldsymbol{\varepsilon}). \quad (23)$$

According to (22) and (23), we first have

$$\left\|\widehat{\boldsymbol{A}} - \breve{\boldsymbol{A}}\right\|_F$$

$$= \left\|\text{vec}\left[\frac{1}{n}\sum_{i=1}^{n}Y_i S(\boldsymbol{X}_i)\right] - \text{vec}\left[\frac{1}{n}\sum_{i=1}^{n}Y_i \hat{S}(\boldsymbol{X}_i)\right]\right\|_2$$

$$= \left\|\left(\boldsymbol{\Sigma}^{-1}\widehat{\boldsymbol{\Sigma}} - \boldsymbol{I}\right)\sum_{r=1}^{R}\left(\boldsymbol{b}_r \otimes \nabla f(\boldsymbol{0}_{p\times R})_{[:,r]}\right) - \frac{1}{n}\left(\boldsymbol{\Sigma}^{-1} - \widehat{\boldsymbol{\Sigma}}^{-1}\right)\boldsymbol{X}^{\top}(\boldsymbol{R}+\boldsymbol{\varepsilon})\right\|_2$$

$$\leqslant \underbrace{\left\|\boldsymbol{\Sigma}^{-1}\widehat{\boldsymbol{\Sigma}} - \boldsymbol{I}\right\|_2\left\|\sum_{r=1}^{R}\left(\boldsymbol{b}_r \otimes \nabla f(\boldsymbol{0}_{p\times R})_{[:,r]}\right)\right\|_2}_{A1} + \underbrace{\frac{1}{n}\left\|\boldsymbol{\Sigma}^{-1} - \widehat{\boldsymbol{\Sigma}}^{-1}\right\|_2\left\|\boldsymbol{X}^{\top}(\boldsymbol{R}+\boldsymbol{\varepsilon})\right\|_2}_{A2}.$$

To bound the target error, we shall bound the A1 and A2 terms respectively. For the first term A1, it holds that

$$\left\|\boldsymbol{\Sigma}^{-1}\widehat{\boldsymbol{\Sigma}} - \boldsymbol{I}\right\|_2 = \left\|\boldsymbol{\Sigma}^{-1}\left(\widehat{\boldsymbol{\Sigma}} - \boldsymbol{\Sigma}\right)\right\|_2 \leqslant \frac{1}{\lambda_m}\left\|\widehat{\boldsymbol{\Sigma}} - \boldsymbol{\Sigma}\right\|_2.$$

According to Lemma D.7,

$$\left\|\widehat{\boldsymbol{\Sigma}} - \boldsymbol{\Sigma}\right\|_2 \leqslant e, \text{ where } e = \max(\delta, \delta^2) \text{ and } \delta = CLt\sqrt{\frac{pd}{n}}$$

with probability at least $1 - 2\exp(-t^2 pd)$. Here $L = \max(K, K^2)$ and $K = M\sqrt{\lambda_M}$. Let $t = \sqrt{\frac{\log(2/\alpha)}{pd}}$, we have

$$\mathbb{P}\left(\left\|\widehat{\boldsymbol{\Sigma}} - \boldsymbol{\Sigma}\right\|_2 \geqslant CL\sqrt{\frac{\log(2/\alpha)}{n}}\right) \leqslant \alpha,$$

which means that

$$\left\|\widehat{\boldsymbol{\Sigma}} - \boldsymbol{\Sigma}\right\|_2 = \mathcal{O}_p\left(\sqrt{1/n}\right). \quad (24)$$

Besides, note that

$$\left\|\sum_{r=1}^{R}\left(\boldsymbol{b}_r \otimes \nabla f(\boldsymbol{0}_{p\times R})_{[:,r]}\right)\right\|_2 = \left\|\text{vec}\left(\nabla f(\boldsymbol{0}_{p\times R})\boldsymbol{\Theta}^{\top}\right)\right\|_2$$

$$= \left\|\nabla f(\boldsymbol{0}_{p\times R})\boldsymbol{\Theta}^{\top}\right\|_F$$

$$= O(\sqrt{p}). \quad (25)$$

Therefore, we get the order of A1 is $\mathcal{O}_p\left(\sqrt{p/n}\right)$. For the second term A2, according to Lemma D.9, we first have

$$\left\|\boldsymbol{\Sigma}^{-1} - \widehat{\boldsymbol{\Sigma}}^{-1}\right\|_2 \leqslant \left\|\boldsymbol{\Sigma}^{-1}\right\|_2\left\|\widehat{\boldsymbol{\Sigma}}^{-1}\right\|_2\left\|\widehat{\boldsymbol{\Sigma}} - \boldsymbol{\Sigma}\right\|_2,$$

where for $\left\|\widehat{\boldsymbol{\Sigma}}^{-1}\right\|_2$ we have $\left\|\widehat{\boldsymbol{\Sigma}}^{-1}\right\|_2 = \lambda_{\max}\left(\widehat{\boldsymbol{\Sigma}}^{-1}\right) = 1/\lambda_{\min}\left(\widehat{\boldsymbol{\Sigma}}\right) = 1/\sigma_{\min}^2(\boldsymbol{X}/\sqrt{n})$. According to Lemma D.8, one has

$$\sigma_{\min}(\boldsymbol{X}/\sqrt{n}) \geqslant 1 - tL\sqrt{pd/n}$$

with probability $1 - 2pd\exp(-ct^2)$. Let $t = \sqrt{\frac{1}{c}\log\left(\frac{2pd}{\alpha}\right)}$ and we have

$$\mathbb{P}\left(\sigma_{\min}(\boldsymbol{X}/\sqrt{n}) \geqslant 1 - L\sqrt{\frac{pd}{cn}\log\left(\frac{2pd}{\alpha}\right)}\right) \leqslant \alpha.$$

Therefore we can get the order of $\sigma_{\min}(\boldsymbol{X}/\sqrt{n})$ is $\mathcal{O}_p(1)$, and then

$$\left\|\widehat{\boldsymbol{\Sigma}}^{-1}\right\|_2 = 1/\sigma_{\min}^2(\boldsymbol{X}/\sqrt{n}) = \mathcal{O}_p(1). \tag{26}$$

Consequently,

$$\left\|\boldsymbol{\Sigma}^{-1} - \widehat{\boldsymbol{\Sigma}}^{-1}\right\|_2 = \mathcal{O}_p\left(\sqrt{1/n}\right). \tag{27}$$

Besides, for the noise-related term $(1/n)\left\|\boldsymbol{X}^\top(\boldsymbol{R} + \boldsymbol{\varepsilon})\right\|_2 \leqslant (1/n)\|\boldsymbol{X}\|_2\left(\|\boldsymbol{R}\|_2 + \|\boldsymbol{\varepsilon}\|_2\right)$. Again by Lemma D.8, we get $\sigma_{\max}(\boldsymbol{X}/\sqrt{n}) = \mathcal{O}_p(1)$, implying that $\|\boldsymbol{X}\|_2 = \mathcal{O}_p(\sqrt{n})$. For the remainder vector $\boldsymbol{R}$, we have

$$h(\boldsymbol{X}_i\boldsymbol{\Theta})\|\boldsymbol{X}_i\boldsymbol{\Theta}\| \leqslant H\|\boldsymbol{X}_i\|_2.$$

Besides, according to Lemma D.10 one has

$$\|\boldsymbol{X}_i\|_2 \leqslant \|\boldsymbol{X}_i\|_F = \|\boldsymbol{x}_i\|_2 = \mathcal{O}_p(\sqrt{pd}).$$

Therefore

$$\|\boldsymbol{R}\|_2 = \sqrt{\sum_{i=1}^n \left(h(\boldsymbol{X}_i\boldsymbol{\Theta})\|\boldsymbol{X}_i\boldsymbol{\Theta}\|\right)^2} \leqslant H\sqrt{\sum_{i=1}^n \|\boldsymbol{X}_i\|_2^2} = \mathcal{O}_p(\sqrt{npd}).$$

For sub-Gaussian $\boldsymbol{\varepsilon}$, according Lemma D.10 the order of $\boldsymbol{\varepsilon}$ is $\mathcal{O}_p(\sqrt{n})$. Therefore the order of noise-related term is

$$\frac{1}{n}\left\|\boldsymbol{X}^\top(\boldsymbol{R} + \boldsymbol{\varepsilon})\right\|_2 = \mathcal{O}_p(\sqrt{pd}). \tag{28}$$

Therefore the order of A2 is $\mathcal{O}_p\left(\sqrt{pd/n}\right)$.

In summary, according to (24) to (28), we can get

$$\left\|\widehat{\boldsymbol{A}} - \breve{\boldsymbol{A}}\right\|_F = \mathcal{O}_p\left(\sqrt{\frac{pd}{n}}\right).$$

Then we can get by Theorem 19 in O'Rourke et al. (2018) again

$$\|\breve{\boldsymbol{\Theta}}\breve{\boldsymbol{O}} - \widehat{\boldsymbol{\Theta}}\|_F \leqslant \sqrt{2}\sin\Theta(\breve{\boldsymbol{\Theta}}, \widehat{\boldsymbol{\Theta}}) \leqslant \frac{2^{3/2}\|\breve{\boldsymbol{A}} - \widehat{\boldsymbol{A}}\|_F}{\sigma_R} = \mathcal{O}_p\left(\sqrt{\frac{Rd}{n}}\right).$$

This completes the proof. $\qquad\square$

