# OpenReview forum: "Learning Convolutional Representations via Generalized Stein’s Method: A Training-Free Approach"
_ICLR.cc/2026/Conference — ICLR 2026 Conference Withdrawn Submission_

### Official Review · Reviewer_3Ggw · 2025-10-25

**Soundness:** 1
**Presentation:** 3
**Contribution:** 2
**Rating:** 2
**Confidence:** 4

**Summary:**

The authors introduce a new algorithm for learning convolutional layer parameters outside of gradient descent, based on SVD and Stein's method. They evaluate it on simulation experiments and on Alzheimer's disease prediction from brain MRI.

**Strengths:**

1. A new approach at representation learning outside of gradient descent is quite interesting.
2. The theoretical motivation of the approach seems grounded, and the interpretation of a convolutional layer as a linking function with the filters being index vectors is novel to my knowledge. All of the theoretical results appear correct.
3. The paper is generally well-written.
4. The authors provided extensive code and a worked example.

**Weaknesses:**

**Major Weaknesses:**
1. **My biggest concern of this paper is the crucial lack of support for relevance to modern deep learning and computer vision.** All experiments are performed on extremely simple networks (2 layers), on very low resolution images (28x28 or 48x48), and there is an overreliance on the results of simple simulations (section 5). The authors state in the abstract that a motivation of their work is the challenge and computational expense of training CNNs in high dimensional, low sample settings, yet to my knowledge (and experience), training CNNs via gradient descent have been remarkably powerful and stable over many years, over a range of data dimensionalities and sample sizes. Moreover, as mentioned the authors only test on very low-dimensional input data, nor do they evaluate the computational cost of their method compared to gradient descent. It is completely unclear how the proposed method scales up to higher dimensional images (e.g., ImageNet and beyond) or CNN architectures (e.g., ResNet, ConvNext, etc.), both in terms of its ability to learn useful representations, and its computational costs.
2. Continuing on this: another issue is how the proposed method is compared to the state-of-the-art (in addition to the aforementioned issues of unrealistically small "toy" models and data dimensionalities). For the simulation experiments of section 5, the evaluation metric (column distance) just evaluates whether their method's parameter updates converge to some simulated target parameters. However, this provides no obvious support of such behavior when training on real data/converging to parameters learned from real data.
3. Even when the authors do turn to evaluating on a real dataset (brain MRI for predicting Alzheimer's disease MMSE), as mentioned the images are again low-dimensional and the network quite simple. But moreover, the proposed method's MMSE result is only compared to other versions of the method, not compared to any state-of-the-art CNN architectures (or even a simple one like ResNet-18), basically resulting in just an ablation study. There is no clear indication from this that their method is useful and relevant for this medical image analysis task (or others) compared to the state-of-the-art.

**Minor Weaknesses:**
In the "Notations" section, everything after the description of the vectorization and inverse operations can be skipped, as these are all standard notation.

**Questions:**

Above Fig 1 it says "results are based on 100 independent repetitions". Assuming this means that the authors ran 100 trials, why don't the results in Fig. 1 (and other figures) have error bars/confidence intervals?

---

### Official Review · Reviewer_ysf1 · 2025-10-30

**Soundness:** 3
**Presentation:** 1
**Contribution:** 2
**Rating:** 2
**Confidence:** 3

**Summary:**

This paper connects the model
$ y = f(Bx) $
where B are filters B:R^N -> R^M and f:R^M ->R
and show that the model can be learned by the Stein’s identity.

I find the paper interesting on the Theoretical size but difficult to follow.

**Strengths:**

As much as I know, this paper contains the first theoretical framework for training-free CNN representation learning via Stein’s method and it bridges classical statistics (index models, Stein identities) with modern deep learning.

Furthermore, the paper provides proofs, with some experiments.

**Weaknesses:**

I have two main problems with this paper
1. The main problem is clarity. Since most of the interesting points of this paper are theoretical it is not reasonable to leave all the real math to the appendix. If the focus is the connection of the model and in particular, the training through Stein’s method, then I expect to get a very clear picture of the connection. This requires a major re-write of the paper which I do not believe that the conference allows

2. I believe that the theoretical restriction of the method are hard. You limit yourself to the first layer of the CNN
3. You assumes knowledge (or parametric estimation) of the input distribution to compute the score function

**Questions:**

See above

---

### Official Review · Reviewer_5b6e · 2025-10-31

**Soundness:** 2
**Presentation:** 3
**Contribution:** 2
**Rating:** 2
**Confidence:** 3

**Summary:**

This paper provides a new method for learning convolutional filters. Under some assumptions, the authors consider the first layer of CNN as an index model and propose an approach for estimation of its parameters using Stein’s formula and SVD. The authors provide theoretical guarantees on the convergence of the estimator. The study on samples from Gaussian distribution and experiments on medical images show competitive or superior performance compared to trained CNNs, especially in small-sample regimes.

**Strengths:**

The paper is well-written and easy to follow. The core idea is novel. The authors provide rigorous theoretical analysis of the proposed method. The experiments cover different input distributions, link functions and medical imaging datasets.

**Weaknesses:**

1. The proposed approach requires the knowledge of (or some assumptions on) the score function of the data, which is the main limitation. Throughout the work, the authors assume that the data follows some distribution with known score function (e.g. Gaussian, Student, Gamma) or score function can be easily estimated, which in general is not the case.
2. In addition, this approach requires training a neural network on top of the extracted features. Therefore, the relative gain compared to traditional gradient descent is unclear.
3. I doubt that the distance metric provided in section (11) is good for comparison of proposed approach to the gradient descent (Adam). As a link function f, the FCN with batchnorm is used, which is invariant to the multiplication of preceding weights by a positive constant. So, if we multiply $\Theta$ by some constant $c>0$, the output $y$ would not change. It means that gradient descent may learn $c\Theta$ instead of $\Theta$, which is correct, but may produce a higher distance metric. It may be more appropriate to normalize $\Theta$ obtained by the gradient descent to have the same norm before measuring the distance.
4. In the case with many filters, the method recovers $\Theta$ only up to an orthogonal transformation.

**Questions:**

1. Modern CNNs consist of multiple convolutional layers, which are stacked one after the other and alternated with nonlinear activation functions. Is it possible to train multiple layers with the proposed approach? Is it computationally and statistically feasible?
2. How could non-parametric high-dimensional distributions (e.g. ImageNet images) be handled?

---

### Official Review · Reviewer_B9x9 · 2025-11-02

**Soundness:** 2
**Presentation:** 2
**Contribution:** 3
**Rating:** 4
**Confidence:** 3

**Summary:**

The paper presents a method to learn convolution filters on inputs to be passed to general nonlinear link functions for downstream use.  By using a matrix formulation of a multi-index model, Stein's method can be used to create an estimator of the of these initial convolution filters through an SVD calculation.  Theoretical results are provided on the sample complexity of the method.  Experiments show that under toy scenarios the true filters of a model can be recovered more effectively than with purely gradient based methods.  Further experiments claim that when used for learning the response variable for a regression model in the low data regime, the proposed method performs better than fully gradient based methods in terms of response variable error as well.

**Strengths:**

The development of methods for relatively low sample scenarios, such as in medical applications, is an important area of research and this work makes a contribution in this direction.

The theoretical treatment of the method is sound and generally well presented.

**Weaknesses:**

The differences in the matrix formulation and the standard MIM feel slightly exaggerated.  For example, in line 158-159 it is claimed that the resulting matrix formulation is a more general function with an input dimension of $p1_1p_2R$ than the typical MIM.  However, the appropriate vectorization of the set of arguments in the link function in (7) would reduce this form to that of a standard MIM.  Specifically, if we reshape $(Xb_1, \ldots, Xb_R) \to ((Xb_1)^T, \ldots, (Xb_R)^T) \in \mathbb{R}^{p_1p_2R}$, we can see (7) as a MIM on $p_1p_2R$ scalar inputs, each of which can be written as a linear function of $\mathrm{vec}(X)$.  The benefit of this matrix formulation comes in the next section when examining the matrix form of Stein's identity which motivates using the spectral properties of the resulting objects; in my opinion this is what should be emphasized instead.

In section 6 there are some details missing around the setup of the baseline method for comparison.  How are the networks in the link functions initialized / trained / are they trained at all?  It seems that the proposed method is mainly being used as an estimator for a good initial representation of the data to be inserted to a downstream model, such as the linear, FCN, or CNN.  It is claimed that this experiment validates its effectiveness as a dimension reduction method for imaging data.  However, there are no other dimensionality reduction methods compared against here.  It would greatly improve the quality of the paper to make this comparison more complete.

Lastly, the code in the supplementary material appears incomplete.  It appears to only contain code for fitting the initial convolution filters with the proposed method for the synthetic experiments.  There is no code showing how the baseline methods for comparison were implemented (details regarding the parametric optimization alternatives, e.g ADAM).  Additionally there is no code for the experiments around measuring errors for response variables in the downstream regression tasks.

**Questions:**

1. How does effectiveness of method scale with increasing complexity of link function i.e. larger FCNs or CNNS.

2. When using the proposed method for predicting response variables, is the link function ever updated?  Or is it randomly initialized and never modified, using only the change of convolution filters on the inputs for fitting the regression model?

3. How would this method compare against other dimensionality reduction techniques other than fully parametric gradient based methods?

---

### Note · Authors · 2025-11-17

I have read and agree with the venue's withdrawal policy on behalf of myself and my co-authors.